# Efficient and Generalizable Mixed-Precision Quantization via Topological Entropy

**Nan Li**
School of Artificial Intelligence, Institute of Big Data Science and Industry
Shanxi University
College of Software, Northeastern University
linan10@sxu.edu.cn

**Yonghui Su**
College of Software,
Northeastern University
2371400@stu.neu.edu.cn

**Lianbo Ma**[*]
College of Software,
Northeastern University
malb@swc.neu.edu.cn

## Abstract

Network quantization effectively reduces both memory footprints and inference time of deep neural networks, enabling their deployment on resource-constrained devices. To fully utilize the multiple bit-width arithmetic operations of the hardware, mixed-precision quantization (MPQ) is developed to assign different bit-widths to each layer. However, the quantization policy obtained by existing MPQ methods struggles to achieve the objectives of efficiency and generalization simultaneously. In this paper, we propose an efficient and generalizable MPQ based on topological entropy (TE) (GMPQ-TE). Specifically, TE, derived from *topological data analysis*, effectively measures the quantization sensitivity of each layer by using the minibatch of data with the same label. Furthermore, we observe that TE remains consistent across various datasets and shows a strong correlation with both quantized model accuracy and bit-width. Thus, MPQ is formulated as a single-pass linear programming problem, obtaining a generalizable quantization policy in a few seconds (11s on MobileNet-V2). Extensive experiments show that the quantization policy obtained on CIFAR-10 can generalize to ImageNet and PASCAL VOC. GMPQ-TE achieves a competitive accuracy-complexity trade-off compared to state-of-the-art MPQ methods.

## 1 Introduction

Deep neural networks have gained increasing attention in image classification Sandler et al. [2018], He et al. [2016], semantic segmentation Strudel et al. [2021], Li et al. [2022], object detection Wang et al. [2019a], Zou et al. [2023], and other vision tasks Xu et al. [2022], Li et al. [2024]. However, due to their extremely high complexity, it is impractical to directly deploy on mobile devices with limited battery capacity and computational resources Wang et al. [2021]. Therefore, there is a need for the model compression method based on the given hardware configurations. Recently, various compression methods have been developed to reduce the model complexity, such as pruning He et al. [2017], knowledge distillation Gou et al. [2021], quantization Wang et al. [2021], Koryakovskiy et al. [2023], and compact architecture design Sandler et al. [2018], Jiang et al. [2023]. Among these methods, quantization aims at mapping weight or activation to lower bit-width for compression and

---

[*]Corresponding author.

acceleration Wang et al. [2021]. To fully utilize arithmetic operations with variable bit-width in hardware platforms, mixed-precision quantization (MPQ) is presented to configure bit-width for each layer, achieving a better trade-off between complexity and accuracy Ma et al. [2023].

However, conventional MPQ methods Wang et al. [2019a], Cai and Vasconcelos [2020] are subject to the limitations of achieving both efficiency and generalization simultaneously. This is because that 1): It is commonly an iterative search in which candidate quantization policies in each generation are required to train and evaluate on the given datasets Chen et al. [2019], Sun et al. [2022a]; 2) Existing approaches generally depend on the given datasets to search for the optimal quantization policy, which cannot be generalized across various datasets due to differences in the distribution of various datasets Cai and Vasconcelos [2020], Wang et al. [2024]. The quantization policy needs to be re-searched on the rare or large-scale datasets Deng et al. [2009], posing significant computational and generalization challenges.

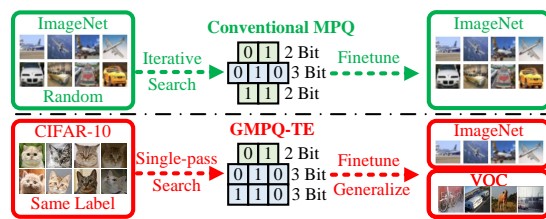

Figure 1: Conventional MPQ methods require the consistency of datasets between bit-width search and model deployment, while GMPQ-TE obtains the optimal quantization policy by using the minibatch of data with the same label and generalizes it to other datasets.

Accordingly, we design an efficient and generalizable MPQ via topological entropy (TE) (GMPQ-TE). Different from the existing approaches that require iteration and cannot generalize to various datasets, GMPQ-TE only requires a single-pass search process on a minibatch of data with the same label to compute the TE and obtain the quantization policy that can be generalized across various datasets (see Figure 1). Specifically, TE measures the stability of the representation (i.e., the quantization sensitivity) of a layer or a network model concerning common features in a

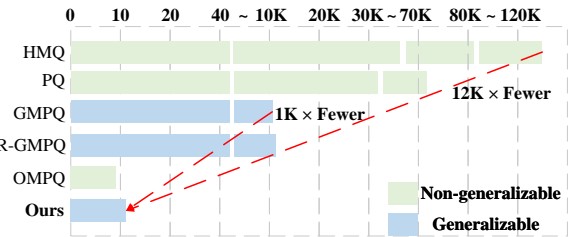

Figure 2: Comparison of the search cost used to obtain the optimal quantization policy on MobileNet-V2 between GMPQ-TE and other MPQ methods.

minibatch of data with the same label. Thus, TE possesses superior cross-dataset consistency, which ensures that the obtained quantization policy can be generalized to other datasets. Furthermore, we observe that: 1) Model TE negatively correlates with quantization performance across different networks. 2) TE of layer is positively correlated with bit-width. Based on this, we minimize the model TE as an objective function and construct a linear programming problem, where a larger bit-width is assigned to the layer with larger TE under specific hardware constraints. Thus, the optimal quantization policy can be obtained by solving the linear programming problem. Our contributions can be listed as follows:

- We first attempt to use TE for measuring the quantization sensitivity of a network model or a layer by using a minibatch of data with the same label. The quantization policy solved by the TE can efficiently be generalized across various datasets. Based on two positive correlations, a single-pass linear programming is designed for MPQ. Such linear programming is solved in a few seconds.

- We provide a comprehensive theoretical analysis of the performance degradation boundary and resolution– and label–independent of TE. Furthermore, we theoretically prove that it is feasible to integrate TE into quantization-aware training.

- We conduct extensive experimental results on image classification and object detection, which show that the quantization policy generalized from CIFAR-10 to ImageNet and PASCAL VOC achieves a competitive accuracy-complexity trade-off compared with the state-of-the-art MPQ methods. Additional real-world deployment results on diverse hardware platforms could further verify the advantages of GMPQ-TE.

## 2  Background Knowledge

**Mixed-Precision Quantization.** Prior MPQ methods are a search-based approach limited by expensive quantization policy evaluations and iterative search process Wang et al. [2019a], Cai and Vasconcelos [2020]. For example, HAQ Wang et al. [2019b] performs 600 quantization policy evaluations. Most recently, GMPQ Wang et al. [2021] and its extended version R-GMPQ Wang et al. [2024] design the generalizable framework via the attribution rank preservation, which focuses on the contribution of each input component to network output. The experimental results prove that it possesses good consistency across various datasets. Thus, the quantization policy searched by GMPQ and R-GMPQ can be generalized across multiple datasets. However, it still requires massive computational overhead (2.8 GPU hours).

To reduce the searching cost, several works develop some "critics" to judge the quantization sensitivity of the layer, such as Hessian eigenvalue Dong et al. [2019], orthogonality Ma et al. [2023], quantization entropy Sun et al. [2022b], first-order information Chauhan et al. [2023], and layer-wise importance Tang et al. [2022]. These metrics often require a random minibatch of data to measure the sensitivity of the layer. Among them, OMPQ Ma et al. [2023] and LIMPQ Tang et al. [2022] define MPQ as a single-pass linear programming problem, which is solved in less than a second. Especially OMPQ requires only 64 images from ImageNet to calculate the optimal quantization policy for ResNet-18/50. However, due to the significant differences in the distribution and resolution of randomly selected data across different datasets, the quantization policy fails to generalize effectively to other datasets. Additionally, this type of linear programming approach, except for OMPQ, operates in the post-training quantization (PTQ) and does not integrate with quantization-aware training (QAT).

In contrast, GMPQ-TE addresses both search efficiency and generalization. This is because TE focuses on the ability of a model or layer to preserve common features of data with the same label rather than their ability to represent random data, eliminating the effects of the obvious difference in the distribution of random data from different datasets. Also, **TE can be integrated into QAT and proven theoretically**(see App. C.1). **The theoretical analysis (see App. C.2) demonstrate the resolution- and label-independent nature of TE.** Specifically, data with different labels from different datasets do not affect the evaluation of quantization sensitivity. Moreover, the single-pass linear programming problem, built upon the property of TE, is solved in under a few seconds.

**Topological Data Analysis.** It aims to explore relationships between neural network and input data Ghrist [2008] based on the key insight "*data has shape*" Carlsson [2014]. Typically, natural images with the same label share common features and the locations of these features are spatially correlated globally. Effective functions highlight these features with high activation values, preserving the global spatial pattern. Conversely, ineffective functions fail to represent these features effectively, resulting in chaotic and vague representations. Based on this observation, recent works have attempted to use topological data analysis to explain the effect of network Rieck et al. [2019], Gabrielsson and Carlsson [2019], Guss and Salakhutdinov [2018], Hofer et al. [2017] or basic operators (e.g., ReLu Naitzat et al. [2020] function Zhao and Zhang [2022]) on performance. For example, Rieck et al. Gabrielsson and Carlsson [2019] design neural persistence based on a topological complexity of network structure, which can provide criteria for early stopping. Guss et al. Guss and Salakhutdinov [2018] empirically provide the correlation between neural network expressiveness and the topological complexity of dataset. Zhao et al. Zhao and Zhang [2022] construct feature entropy to evaluate the effectiveness of functions via topological data analysis. Compared to the above studies, TE based on the observation from work Zhao and Zhang [2022] can finely evaluate functions from the channel level. Furthermore, we pioneer to discover its relationship with quantized model accuracy and confirm its validity in MPQ from the view of TE.

**Key Concept.** Here, we mainly introduce the concepts from graph theory used in constructing TE.

*Clique.* It refers to a complete subgraph from a graph, where a clique is a subset of nodes in the graph and every pair of nodes in this subset is mutually connected Carlsson [2014].

*Clique Filtration.* Given a graph, a filtration is constructed by gradually adding cliques (complete subgraphs) during the filtration process. Each clique corresponds to a threshold at which it forms in the graph, and additional cliques are added as the threshold changes. This filtration process helps analyze the evolving topological structures in the graph as the threshold varies Guss and Salakhutdinov [2018].

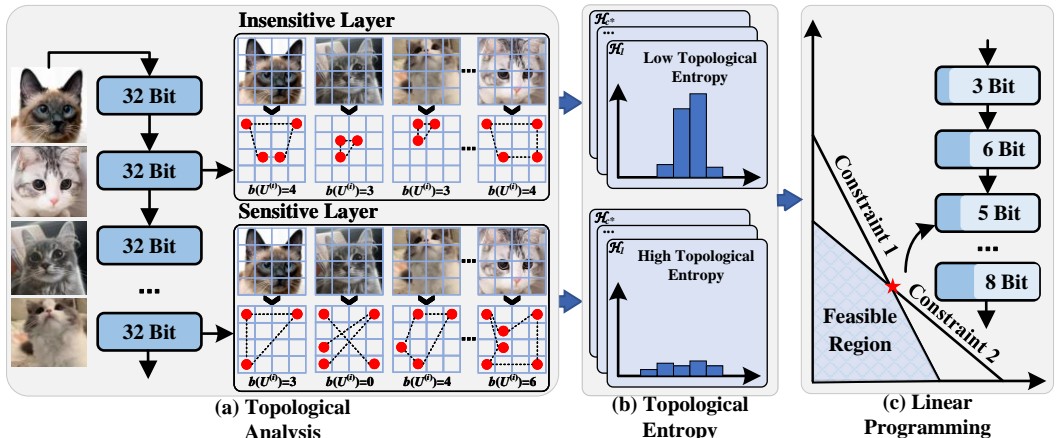

Figure 3: The overall framework of the proposed GMPQ-TE method. (a) Analyze a minibatch of data with the same label and obtain Betti time. (b) Calculate TE based on Betti time distribution. (c) Linear programming problem constructed by TE to derive optimal generalizable quantization policy.

*Persistent Homology.* It is used to capture how the data evolves in terms of its structure at various scales. Specifically, it starts by tracking simple "starting points" in the data and gradually adds components (e.g., cliques), observing how these topological features "form" and "disappear" Zhao and Zhang [2022]. By calculating the homology of the data, persistent homology identifies topological features (such as connected components, loops, voids, etc.) and provides insights into their birth and death across different scales. This is especially useful in understanding complex data structures.

*Betti Curve.* It is a significant criterion of persistent homology. It provides a visualization of the "persistence" of topological features in data across different dimensions Zhao and Zhang [2022]. The Betti curve depicts the evolution of these values (for example, how Betti numbers change through filtration) and helps us understand how topological features evolve across different scales of the data, where Betti numbers describe the number of topological features in each dimension (e.g., the 1-st Betti number is equal to the number of circles in the graph).

*Birth Time.* For each topological feature (e.g., circle), the birth time refers to the time or scale at which the feature first forms. It marks the initiation of the feature's existence in the data. It is crucial for capturing the fundamental structure of the data Bochner [1948].

## 3 Methodology

### 3.1 Topological Analysis

As shown in Figure 3, given the input data with the same label $\mathcal{X} \in \mathbb{R}^{c \times h \times w}$, a neural network with $L$-layers can be considered as an input-to-output mapping function consisting of $L$ functions. The output feature maps of each function are $\mathcal{O} \in \mathbb{R}^{c^* \times h^* \times w^*}$, which can be unfolded along the channel dimension to obtain $c^*$ feature maps ($\mathcal{F} = \{f^{(1)}, f^{(2)}, ..., f^{(c^*)}\}$). Each feature map $f^{(i)} \in \mathbb{R}^{h^* \times w^*}$[2] can be viewed as a grid structure $U^{(i)} \in \mathbb{R}^{h^* \times w^*}$. Intuitively, the spatial patterns within the elements reveal specific regular relationships among their coordinate indices. It is natural to capture such relationship using graph structure and tackle it through topological analysis.

**Graph Construction.** We construct the edge-weighted graph $\mathcal{G} = (V, E)$ for the grid structure $U^{(i)}$, where $V$ and $E$ are the vertex set and the edge set, respectively. The weighted adjacency matrix $\mathbf{W}$ of $\mathcal{G}$ is defined as $\mathbf{W} \in \mathbb{R}^{h^* \times w^*} : W_{j,k} = U^{(i)}_{j,k}$, where $(j, k)$ denotes the coordinate index of the location of an element in $\mathbf{W}$ and $U^{(i)}$. Note that each element of $\mathbf{W}$ is the weight of edge in $\mathcal{G}$, which conveys the intensity of the corresponding point in the $U^{(i)}$. Based on the typical implementation of

---

[2]We provide construction method across different architectures in App. A.

the sublevel set (see Eq. 1), we can construct the undirected subgraphs $\mathcal{G}^{(v)} = (V^{(v)}, E^{(v)})$, where $V^{(v)} = V$ and $E^{(v)} \subset E$. Its weighted adjacency matrix is denoted as $\mathbf{W}^{(v)}$.

$$\mathbf{W}_{j,k}^{(v)} = \mathbf{I}(\mathbf{W}_{j,k} \geq w^{(v)}) \tag{1}$$

where $\mathbf{I}(.)$ is indicator function. $w^{(v)}$ is the $v$-th value in the descend ordering of elements of $\mathbf{W}^{(v)}$. If $\mathbf{W}_{j,k}$ is larger than or equal to $w^{(v)}$, then $\mathbf{W}_{j,k}^{(v)}$ equals 1, and conversely, it equals 0. Graph filtration requires to construct subgraphs based on the weights of the edges. If the weights of the edges are not symmetric, it may lead to some unreasonable graph structures, such as wrong edge connections or unidirectional connections Zhao and Zhang [2022]. To ensure that $\mathbf{W}_{j,k}^{(v)}$ is symmetric, we make the following adjustment.

$$\mathbf{W}^{(v)} = \max(\mathbf{W}^{(v)}, (\mathbf{W}^{(v)^T})) \tag{2}$$

By utilizing Eq. 2, we obtain the graph filtration $\mathcal{G}_{(1)} \subset \mathcal{G}_{(2)} \subset \cdots \subset \mathcal{G}$. In this sublevel set filtration (i.e., $\mathcal{G}_i$), the process commences with the vertex set. The edge weights are then ranked from the maximum $w_{\max}$ to the minimum $w_{\min}$, and the threshold parameters are systematically decreased from $w_{\max}$ to $w_{\min}$. At each step, the corresponding edges are incorporated to form the threshold subgraph $\mathcal{G}^{(v)}$.

**Clique Filtration.** To further explore the structural information among the elements in the threshold subgraphs, we use topological invariants to capture high-level abstraction of structural information. Here, each subgraph $\mathcal{G}^{(v)}$ is converted to clique complex $\mathcal{K}_v$ based on persistent homology method Horak et al. [2009]. In this way, we can get clique complex filtration ($\mathcal{K}_1 \subset \mathcal{K}_2 \subset \cdots \subset \mathcal{K}$) corresponding to graph filtration.

The clique complex filtration describes the evolution of structural information in graph $\mathcal{G}$ along with the decreasing threshold parameter ($v$).

**Betti Curve and Birth Time.** $k$-th Betti number can be regarded as the number of k-dimensional *circle* or *hole* in complex. Due to the fact that grid structure $U$ is a matrix of $h^* \times w^*$, the complex contains no higher-order structures (e.g., *circle*). Thus, each element ($\mathcal{K}_v$) in clique complex filtration is quantified by 1-st Betti number $\beta_1(\mathcal{K}_v)$ based on persistent homology theory Hatcher [2002], which can characterize the number of *circle* structural information in complex $\mathcal{K}_v$.

$$\mathcal{K}_v \mapsto \beta_1(\mathcal{K}_v) \tag{3}$$

Intuitively, many meaningful patterns in $U^{(i)}$ would lead to the *circle* structure in the clique complex filtration. The number of *circle* is typically used as an important quantitative index for expressing patterns. Hence, the 1-st Betti number $\beta_1(\mathcal{K}_v)$, $v \in \{1, \cdots, h^*\}$ could be arranged into so called 1-st Betti curves $\beta(U^{(i)}, v)$ for $U^{(i)}$. Here, we employ birth time $b(U^{(i)})$ to interpret and extract core characterization of 1-st Betti curves $\beta(U^{(i)}, v)$.

$$b(U^{(i)}) = \inf\{v \mid \beta(U^{(i)}, v) \neq 0\} \tag{4}$$

Birth time refers to the moment when *circle* structure begins to appear (i.e., (Betti number $\neq 0$)) in the clique complex filtration. It signifies essential changes in filtration, indicating that regularized spatial patterns of notable components begin to emerge within $U^{(i)}$. Conversely, when no spatial pattern is detected in the components of $U^{(i)}$, $\beta(U^{(i)}, v)$ remains 0, indicating the absence of a birth time. This situation typically arises when the function fails to effectively represent the images (i.e., the majority of the values in $U^{(i)}$ are equal to 0). **We provide an example about Betti curve and and birth time in App. B**.

### 3.2 Topological Entropy

If one layer is quantization-insensitive, the output feature maps can still perceive common features about a set of images with the same label at lower bit-width. This is, the birth time obtained from

each realization of $U^{(i)}$ should be relatively close. In other words, the performance of a quantization-insensitive layer for a certain label should remain stable across all images with the same label. This is the key idea behind assessing the performance of layer.

**Birth Time Distribution.** Sampling a set of images with the same label can be regarded as statistical experiments. Thus, birth time is a random variable (denoted as $b(m, U^{(i)})$). We construct the probability space $(\Omega, \Sigma, P)$ for $b(m, U^{(i)})$, where the elements in sample space $\Omega$ are the unit $U^{(i)}$ resulted from the images with same label, $\Sigma$ could be set as common discrete $\sigma$-field and probability measure $P$ is uniformly distributed on $\Omega$ since each image has an equal chance to be selected as the input of network model. This probability space satisfies the following probability distribution,

$$P_{U^{(i)}}(x) = P(b(m, U^{(i)}) = x) = \frac{\sum_{j=1}^{\#(\Omega)} \mathbf{I}(b(U^{(i)}) = x)}{\#(\Omega)}, \tag{5}$$

The degree of concentration of $P_{U^{(i)}}(x)$ gives a direct view of the expressiveness of $U^{(i)}$ for the images with the same label. Specifically, if the distribution presents close to a degenerate-like style, it means that the underlying common features of the images with the same label could be *stably* perceived by $U^{(i)}$. On the contrary, the distribution presents close to a uniform-like style when features are perceived almost blindly, indicating that $U^{(i)}$ is invalid for the images with the same label. In summary, the degree of concentration of $P_{U^{(i)}}(x)$ is supposed to be a quantification sensitivity indicator of $U^{(i)}$.

Considering a function with $c^*$ output features, we use the weighted entropy to further measure the quantification sensitivity of output features of the function (called TE $\mathcal{H}$).

$$\mathcal{H} = -\sum_{i=1}^{c^*} \theta_i * \mathcal{H}_i \tag{6}$$

where

$$\mathcal{H}_i = -\sum_{x}^{|\mathcal{X}|} P_{U^{(i)}}(x) \log P_{U^{(i)}}(x), \quad \theta_i = \frac{\mathcal{H}_i}{\sum_{i=1}^{c^*} \mathcal{H}_i} \tag{7}$$

where $\mathcal{H}_i$ and $w_i$ are the TE and weight corresponding to $U^{(i)}$, respectively. A lower $\mathcal{H}_i$ means that $U^{(i)}$ exhibits quantization-insensitive. Thus, a lower weight $\theta_i$ should be assigned for a lower $\mathcal{H}_i$.

### 3.3 Linear Programming for MPQ

TE directly indicates the quantification sensitivity of the layers (i.e., functions) in the network without taking into account the different dataset distributions, guiding the configuration of a generalizable quantization policy. Generally, sensitive layers should be assigned a larger bit-width to enhance their representational capability Liu et al. [2018]. Thus, we assign a larger bit-width to layers with higher TE to maximize representational capability.

We perform extensive experiments that present sufficient and reliable evidence for such an assertion. Specifically, we first sample various quantization policies for ResNet-18 and Faster R-CNN. We then perform fine-tuning to optimize model performance. Simultaneously, the overall TE of the sampled models is calculated separately. Interestingly, as shown in Figure 4, we find that model TE and performance are positively correlated with the sum of TE of each layer. Based on this finding, we set minimizing TE as our objective function and formulate a linear programming problem to derive final quantization policy.

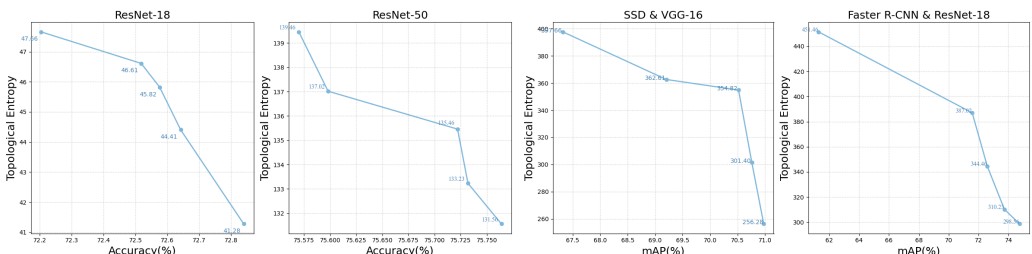

Figure 4: Relationship between TE and accuracy/mAP for different quantization policies on ResNet-18, ResNet-50, SDD & VGG-16 and Faster R-CNN & ResNet-18.

$$\text{Objective} \quad \min_{\mathcal{B}} \sum_{i=1}^{L} \mathcal{H}^{(i)} * \mathcal{B}^{(i)}$$

$$\text{Constraint} \quad \sum_{i=1}^{L} M^{(\mathcal{B}^{(i)})} \leq \mathcal{T}, \quad \mathcal{B}^{(i)} \in \mathbb{N}^{+}, \quad \mathcal{H}^{(i)} > \mathcal{H}^{(j)} \implies \mathcal{B}^{(i)} > \mathcal{B}^{(j)}. \quad (8)$$

where $M^{(\mathcal{B}^{(i)})}$ represents the model size of the $i$-th layer under $\mathcal{B}^{(i)}$ bit-width quantization, while $\mathcal{T}$ denotes the target model size. $\mathcal{B}$ refers to the optimal bit-width configuration, where all $\mathcal{B}^{(i)}$ are integers (i.e., $\mathcal{B}^{(i)} \in \mathbb{N}^{+}$). A larger TE corresponds to a larger bit-width configuration (i.e., $\mathcal{H}^{(i)} > \mathcal{H}^{(j)} \implies \mathcal{B}^{(i)} > \mathcal{B}^{(j)}$). Maximizing the objective function involves assigning a larger bit-width to more unstable layers, thereby implicitly enhancing the representational capacity of the model. Notably, Eq. 8 can be solved in only a few seconds on a single CPU. Furthermore, **we theoretically prove a performance degradation bound for GMPQ-TE (see App. C.3).**

**Remark**: The objective of MPQ is to allocate bit-widths $(B_i \in \mathbb{N}^{+})$ to each layer $(i \in \{1, \ldots, L\})$ under a resource budget $T$, to minimize total performance degradation $\min_{\{B_i\}} \sum_{i=1}^{L} \Delta \mathcal{L}_i(B_i)$, s.t. $\sum_{i=1}^{L} M(B_i) \leq T$, where $\Delta \mathcal{L}_i(B_i)$ denotes the task loss increase due to quantization of layer $i$, and $M(B_i)$ is the corresponding resource cost. We define TE $H_i$ from the birth-time distribution $P_{U^{(i)}}(x)$ of same-label inputs at layer $i$. A high $H_i$ implies structural inconsistency and greater sensitivity to quantization. Quantization introduces perturbations to the weights. For layer $i$, the quantized weight is given by $\hat{W}_i = W_i + \delta W_i, \quad \|\delta W_i\| \leq \varepsilon$.

Assuming a Lipschitz continuous forward operator $f^{(i)}$, the pre- and post-quantization feature maps are $U^{(i)} = f^{(i)}(W_i, X)$, $\hat{U}^{(i)} = f^{(i)}(\hat{W}_i, X)$. By Lipschitz continuity, the output deviation is bounded by $\|U^{(i)} - \hat{U}^{(i)}\| \leq L_i \cdot \|W_i - \hat{W}_i\| \leq L_i \varepsilon$. This perturbation in feature space leads to structural changes in the persistence diagrams, with bottleneck distance bounded by $d_B(D(U^{(i)}), D(\hat{U}^{(i)})) \leq \|U^{(i)} - \hat{U}^{(i)}\|$. Given that entropy is computed over the birth-time histogram derived from the persistence diagram, we can infer the $\ell_1$ difference in their birth-time distributions $\|P_{U^{(i)}} - P_{\hat{U}^{(i)}}\|_1 \leq \alpha_i \cdot \|W_i - \hat{W}_i\| \leq \alpha_i \varepsilon$. Shannon entropy is Lipschitz continuous with respect to $\ell_1$ distance, yielding the following bound $|H_i^{(0)} - H_i^{(B)}| \leq C_i \cdot \|P_{U^{(i)}} - P_{\hat{U}^{(i)}}\|_1 \leq C_i \alpha_i \varepsilon$. We now relate entropy variation to task loss degradation. The increase in loss can be estimated by a second-order Taylor expansion $\Delta \mathcal{L}_i \leq \frac{1}{2} \lambda_i \|\delta W_i\|^2 + \frac{1}{6} \kappa_i \|\delta W_i\|^3$. We further observe that the $L_1$ distance between the birth-time distributions before and after quantization can be upper bounded in terms of TE. Specifically, since Shannon entropy is Lipschitz-continuous with respect to its input distribution, we have $\Phi\left(\|P_{U^{(i)}} - P_{\hat{U}^{(i)}}\|_1\right) \leq \Phi\left(|H_i^{(0)} - H_i^{(B)}|\right)$. This confirms that TE is an upper-bound surrogate of quantization-induced performance degradation. Based on this insight, we propose a TE-guided MPQ objective $\min_{\{B_i\}} \sum_{i=1}^{L} H_i \cdot B_i$ s.t. $\sum_{i=1}^{L} M(B_i) \leq T$. To ensure alignment with sensitivity ranking, we enforce $H_i > H_j \Rightarrow B_i > B_j$. This ordering remains consistent during training due to entropy drift stability, as shown by $|H_i^{(t)} - H_i^{(0)}| \leq \Delta_{q,i} \log M_i \cdot L_i \Rightarrow \text{sign}(H_i - H_j) = \text{sign}(H_i^{(t)} - H_j^{(t)})$, where $\text{sign}(H_i - H_j)$ denotes the relative ordering between layer sensitivities. This ensures that layers identified as more sensitive in the initial phase continue to

receive higher precision during optimization. Thus, TE is suitable for mixed-precision quantization with sufficient theoretical justification.

## 4 Experiments

We first introduce the datasets and the experimental settings. Then, we demonstrate why TE possesses good cross-dataset properties through ablation experiments. Finally, we compare GMPQ-TF to state-of-the-art MPQ approaches for image classification and object detection.

### 4.1 Datasets and Implementation Details

Following studies Wang et al. [2021] and Wang et al. [2024] configuration, for image classification, we use ImageNet Deng et al. [2009] to evaluate the quantized networks, where ResNet-18, ResNet-50 He et al. [2016], MobileNet-V2 Sandler et al. [2018], ViT Dosovitskiy et al. [2020] and Swin transformer Liu et al. [2021] are treated as the baseline architectures. In terms of object detection, PASCAL VOC Everingham et al. [2015] is employed to validate the effectiveness of GMPQ-TE, where VGG-16 Karen [2014] with SSD framework Liu et al. [2016] and ResNet-18 with Faster R-CNN Ren et al. [2016] are

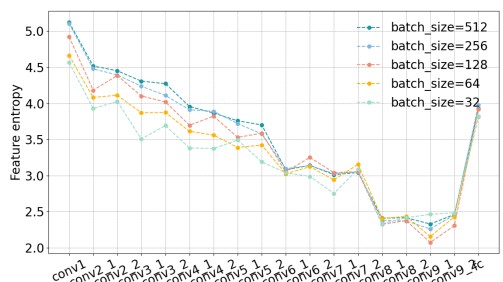

Figure 5: Relationship between different batch sizes and TE on ResNet-18.

used as the baseline architectures. Regarding evaluation metrics, common criteria include model storage cost (Params.), computational cost (BOPs), BOPs compression ratio (Comp), and search cost (s). Top-1/5 accuracy (%) and mean average precision (mAP) (%) Everingham et al. [2015] are specifically used for image classification and object detection, respectively. By setting different constraints $\mathcal{T}$, we can obtain the quantized networks with different accuracy-complexity trade-offs. In addition, we follow study Wang et al. [2021] to fine-tune the quantized networks that are found on different tasks. All experiments are perform on an NVIDIA Geforce GTX 3090Ti.

### 4.2 Ablation Study

**1) Batch size**: Normally, more accurate measures of quantization sensitivity can be achieved by sampling as many images with similar labels as possible. However, excessive data sampling may lead to significant computational overhead. We set different batch sizes (i.e., 32, 64, 128, 256, and 512) on CIFAR-10 to verify the effect of batch size on quantification sensitivity, where ResNet-18 as a baseline model is used to test the TE of each layer. A batch of images with the label "cat" is selected as input for ResNet-18. Figure 5 illustrates the TE of each layer under different batch sizes. Overall, the trend of TE variation is

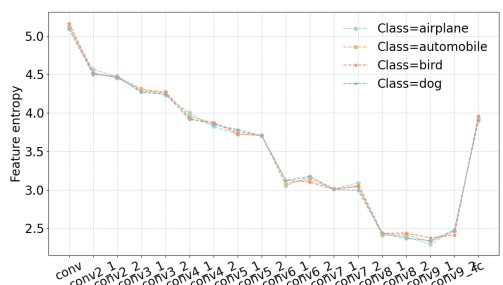

Figure 6: Relationship between different labels from the same dataset and TE on ResNet-18.

consistent across batch sizes. Furthermore, we can observe that the TE of each layer stabilizes as the batch size increases. Since the differences in TE for batch sizes 128, 256, and 512 are insignificant, we consider the computational efficiency and choose 128 in all experiments.

**2) Labels from the different datasets**: We compute TE derived from the labels of different datasets to validate its cross-dataset property. Specifically, we first use ResNet-18 as the baseline architecture and compute TE for each layer obtained from the same labels ("dog" and "cat") of the three datasets (CIFAR-10, ImageNet, PASCAL VOC), as shown in Figure 7 (a). We observe that images with the

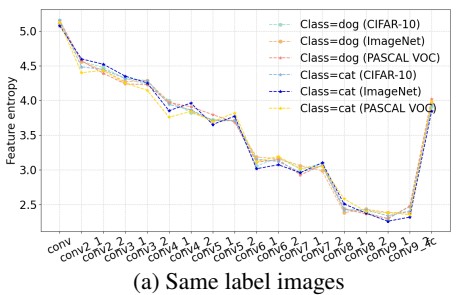
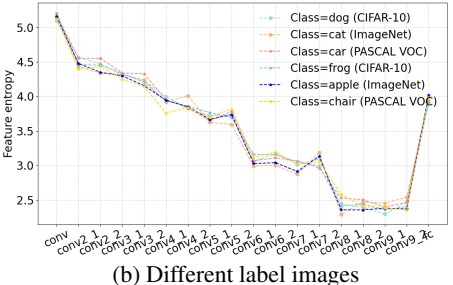

(a) Same label images        (b) Different label images

Figure 7: Relationship between labels from different datasets and TE on ResNet-18.

same labels on different datasets do not affect the calculation of TE. We then use different labels to further check the consistency of TE across datasets, as shown in Figure 7 (b). Similarly, TE is consistent across different labels in different datasets. The above phenomenon indicates that TE has good cross-dataset property. That is, the quantization strategy obtained through TE has good generalization ability.

**3) Labels from the same dataset**: To exclude the effect of labels on TE, we select "airplane", "automobile", "bird", and "dog" in CIFAR-10 for analysis, where ResNet-18 serves as the benchmark architecture. The experimental results are shown in Figure 6. We can observe that TE of each layer does not change dramatically with the change of labels. This proves that the TE is independent of the labels. Therefore, there is no need to consider the effect of labels when using TE for quantitative sensitivity analysis of each layer. This reflects the robustness of the proposed TE concerning labels.

### 4.3 Comparison with State-of-the-art Methods

We compare the proposed method with the state-of-the-art quantization approaches, including OMPQ Ma et al. [2023], GMPQ Wang et al. [2021], R-GMPQ Wang et al. [2024], DJPQ Wang et al. [2020], EdMIDS Cai and Vasconcelos [2020], HAWQ Dong et al. [2019], APoT Li et al., ALQ Qu et al. [2020], HMQ Habi et al. [2020], HAQ Wang et al. [2019a], BP-NAS Yu et al. [2020], RQ Louizos et al. [2020], EMQ Dong et al. [2023], and DQ Uhlich et al. [2019]. There are two types from which these experimental data are derived: duplication of data from the original literature and reproduction based on the open-source code or the quantized architectural information. † is implemented by ourselves using open source code. ∗ is implemented based on the quantized architectural information pro-

Table 1: Results for image classification on MobileNet-V2 (PTQ).

| Methods | Params. | BOPs | Comp. | Top-1 | Top-5 | Cost. |
|---|---|---|---|---|---|---|
| Full-precision | 13.4 | 337.9 | – | 71.9 | 90.3 | – |
| RQ | 2.7 | 11.9 | 28.4 | 68 | – | – |
| GMPQ† | 1.4 | 10.4 | 32.6 | 71.5 | 90.1 | 6.1K |
| R-GMPQ† | 2.2 | 9.9 | 34.1 | 71.7 | 90.2 | 6.8K |
| **GMPQ-TE** | 1.4 | 10.1 | 33.4 | 71.8 | 90.2 | 15 |
| HAQ | 1.4 | 8.3 | 41 | 69.5 | 88.8 | 18.3K |
| DJPQ | 1.9 | 7.9 | 43 | 69.3 | – | 43.9K |
| GMPQ | 1.2 | 7.4 | 45.8 | 70.4 | 89.4 | 9.3K |
| R-GMPQ | 1.1 | 7.2 | 46.7 | 70.9 | 89.7 | 10.4K |
| **GMPQ-TE** | 1.1 | 7.3 | 46.1 | 71.2 | 89.9 | 14 |
| HMQ | 1.7 | 5.2 | 64.4 | 70.9 | – | 120.6K |
| DQ | 1.7 | 4.9 | 68.7 | 69.7 | – | 77.7K |
| EMQ | 1.5 | | | 70.75 | – | few seconds |
| GMPQ† | 1 | 4.8 | 69.7 | 69.5 | 89.1 | 10K |
| R-GMPQ† | 1.3 | 4.7 | 72.2 | 69.7 | 89.5 | 11.1K |
| OMPQ* | 1.5 | 12.3 | 27.3 | 70.39 | 89.9 | 9 |
| **GMPQ-TE** | 1.3 | 4.5 | 74.8 | 69.8 | 89.7 | 11 |

vided by the authors due to lack of open source code. The best and second-best are color coded. The performances of the full-precision baseline models are also provided for better comparison. We use CIFAR-10 to find the suitable quantization policy for all baseline architectures. Then, the baseline architectures with suitable quantization policies are deployed on ImageNet or PASCAL VOC. In addition, the results for **ResNet-18, ResNet-50, ViT, Swin transformer and ResNet-18 with Faster R-CNN under PTQ** can be found in App. D.1. Results under **QAT for ResNet-18, ResNet-50, MobileNet-V2, ViT, Swin transformer** can be found in App. D.2.

**Results on ImageNet:** Table 1 record the experimental results on ImageNet with MobileNet-V2 as the baseline model, respectively. Compared to the generalizable MPQ approaches (i.e., GMPQ and R-GMPQ), GMPQ-TE achieves good post-quantization accuracy over multiple baseline architectures with less cost. Compared to OMPQ, GMPQ-TE gets better quantization performance although GMPQ-TE uses 2s more on MobileNet-V2.

**Results on PASCAL VOC:** As shown in Table 2, GMPQ-TE achieves the best trade-off between mAP and compression ratio on SSD & VGG-16. For example, GMPQ-TE obtains 71.1% mAP with only 32.6 Mb parameters and 758 BOPs, outperforming R-GMPQ in both accuracy and computational efficiency under the same parameter budget. Notably, it also surpasses EdMIPS by 2.2% mAP while using significantly fewer parameters (32.6 Mb vs. 33.5 Mb) and computational cost (27 vs. 5.4K). Meanwhile, the Faster R-CNN & ResNet-18 model compressed by GMPQ-TE yields a 0.7% mAP improvement over GMPQ, while reducing the cost by approximately 250 times, demonstrating its superior efficiency-accuracy balance.

Table 2: Results for object detection on SSD & VGG-16. (PTQ)

| Methods | Params. | BOPs | Comp. | mAP | Cost. |
|---|---|---|---|---|---|
| Full-precision | 105.5 | 27787.7 | – | 72.4 | – |
| HAQ | 42.7 | 847.2 | 32.8 | 70.9 | 225K |
| HAQ-C | 42.9 | 819.7 | 33.9 | 67.6 | 18.3K |
| EdMIPS | 33.5 | 958.2 | 29 | 69.4 | 5.4K |
| GMPQ | 36.6 | 796.2 | 34.9 | 70.5 | 5.7K |
| R-GMPQ | 32.6 | 761.3 | 36.5 | 70.8 | 6.4K |
| **GMPQ-TE** | 32.6 | 758.4 | 36.6 | 71.1 | 27 |
| HAQ | 35.5 | 430.15 | 64.6 | 69.1 | 244.4K |
| HAQ-C | 36.3 | 445.3 | 62.4 | 66.4 | 24.4K |
| EdMIPS | 29.4 | 454 | 61.2 | 68.7 | 108.7K |
| GMPQ† | 24.7 | 413.5 | 67.2 | 69.2 | 6.4K |
| R-GMPQ† | 26.9 | 406.8 | 68.3 | 70.3 | 7.2K |
| **GMPQ-TE** | 24.7 | 405.4 | 68.6 | 70.6 | 23 |

## 4.4 Hardware Efficiency

Table 3 compares the hardware performance of different quantization methods on two FPGA platforms (XC7Z020 and XC7Z045) using ResNet-18 on ImageNet. For each FPGA board, the utilization percentages of look-up table (LUT) and digital signal

Table 3: Comparison of different quantization methods on two FPGA boards for ResNet-18 on ImageNet.

| Methods | Results on FPGA XC7Z020 | | | | Results on FPGA XC7Z045 | | | |
|---|---|---|---|---|---|---|---|---|
| | Utilization | | Throughput | Latency | Utilization | | Throughput | Latency |
| | LUT | DSP | (GOP/s) | (ms) | LUT | DSP | (GOP/s) | (ms) |
| GMPQ | 49% | 100% | 75 | 57.3 | 52% | 100% | 367 | 12.3 |
| EdMIPS | 39% | 100% | 61 | 61.4 | 41% | 100% | 316 | 15.9 |
| R-GMPQ | 50% | 100% | 79 | 52.1 | 59% | 100% | 398 | 10.4 |
| **GMPQ-TE** | 53% | 100% | 84 | 42.3 | 62% | 100% | 413 | 9.4 |

processing (DSP) blocks are reported for each quantization method. The LUT utilization, in particular, reflects the effectiveness of the quantization method in accelerating inference. Among all methods, GMPQ-TE consistently achieves the best overall performance. On XC7Z020, it attains the highest LUT utilization (53%) and throughput (84 GOP/s), while reducing latency to 42.3 ms, significantly lower than GMPQ (57.3 ms), EdMIPS (61.4 ms), and R-GMPQ (52.1 ms). On the more powerful XC7Z045, GMPQ-TE further improves throughput to 413 GOP/s and lowers latency to 9.4 ms, outperforming R-GMPQ (398 GOP/s, 10.4 ms) and GMPQ (367 GOP/s, 12.3 ms). These results demonstrate that GMPQ-TE offers superior hardware efficiency and inference speed compared to existing methods across different FPGA platforms.

## 5 Conclusion

In this paper, we propose GMPQ-TE for handling the efficiency and generalization. By utilizing the properties of TE, GMPQ-TE can be constructed as a single-pass linear programming. The quantization policy obtained on CIFAR-10 is well generalized to ImageNet and PASCAL VOC. The proposed GMPQ-TE outperforms peer MPQ in terms of both accuracy and complexity. Further, the ablation studies verify the effectiveness of the TE. GMPQ-TE offers a promising direction for MPQ. In the future, we will explore new "*topological data analysis*" for MPQ on more complex tasks.

## Acknowledgments

This work is supported in part by the National Natural Science Foundation of China under Grant (No.62472079), in part by the Fundamental Research Funds for the Central Universities (No.N2417003).

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

## A  Feature Map Construction Across Different Architectures

**Case 1 — Convolution-based architectures**. For convolutional networks, the output feature map $\mathbf{U}^{(i)} \in \mathbb{R}^{h^i \times w^i}$ is naturally defined on a regular 2D grid. This grid already encodes the true Euclidean neighborhood structure assumed by our clique filtration process. Therefore, we can directly flatten the $n = h^i \cdot w^i$ vertices to form $\mathbf{A} \in \mathbb{R}^{n \times n}$ without any padding.

**Case 2 — Transformer-based architectures**. Vision Transformers produce a sequence of $N$ tokens, where $N$ is often not a perfect square. Tokens can be rearranged into a $p \times q$ grid according to their spatial positions in the original image, but typically $p \neq q$ and some positions remain empty. A direct flattening of the 1D token sequence into an $N \times N$ adjacency matrix would impose an arbitrary linear ordering on the vertices. This ordering does not correspond to the true Euclidean neighborhood relationships in the image plane, and would distort the clique filtration process by creating spurious edges between spatially unrelated tokens. As a result, the birth–death times of topological features would be inconsistent with those obtained from genuine 2D grids.

To address this, we embed the $p \times q$ token grid into the *smallest enclosing square* of size $\lceil \sqrt{N} \rceil \times \lceil \sqrt{N} \rceil$, and fill the empty positions with zero-valued tokens. This construction has two benefits:

1. It ensures that adjacency is always defined on a square 2D lattice, identical in form to the convolutional case, enabling fair cross-architecture comparison.

2. By choosing the minimal square size, we limit the number of padded vertices, thereby minimizing any dilution of the adjacency structure.

From a topological perspective, the padded region forms a *contractible subcomplex* with trivial higher-order homology. In the filtration, zero-valued vertices outside the original token set do not create new non-trivial cycles. Therefore, persistent homology — and the resulting topological entropy — is invariant under this padding step.

After padding, the feature map has size $h^i \times w^i = \lceil \sqrt{N} \rceil \times \lceil \sqrt{N} \rceil$, and is flattened into an adjacency matrix of shape $(h^i \cdot w^i) \times (h^i \cdot w^i)$. Each matrix entry $\mathbf{A}_{uv}$ encodes the weighted connection between vertices $u$ and $v$ under this consistent 2D spatial embedding.

## B  Betti Curves and Birth Times

We illustrate the concept of Betti curves and birth times using a simple clique filtration process over four nodes labeled 1–4. Let the filtration sequence be denoted as $(\tau^{(1)}, \tau^{(2)}, \ldots, \tau^{(6)})$, where each step incrementally adds edges to build higher-order cliques:

- $\tau^{(1)}$: add edge $(1, 2)$

- $\tau^{(2)}$: add edge $(3, 4)$

- $\tau^{(3)}$: continue adding edges, no cycle yet

- $\tau^{(4)}$: form a 1-cycle $(1 - 2 - 4 - 3 - 1)$

- $\tau^{(5)}$: add edge $(2, 3)$, creating additional cliques

- $\tau^{(6)}$: complete graph with all edges

To capture the emergence of topological features, we define a birth-time indicator function $\beta(i, v, U_k)$, which equals 1 if the $i$-th feature appears for the first time at filtration step $v$, and 0 otherwise. A typical curve looks like $v = [0, 1, 2, 3, 4, 5, 6] \quad \Rightarrow \quad \beta = [0, 0, 0, 0, 1, 0, 0]$

This indicates that a 1-dimensional loop is born at $v = 4$, corresponding to the closure of a cycle. Notably, although edges are added at $v = 5$ and $v = 6$, they do not give rise to new independent topological features. Instead, they reinforce existing structures or create higher-order cliques, thus not being counted as new births.

## C Theoretical Analysis

### C.1 Integration of Quantization-aware Training

Similarly to the study Ma et al. [2023], we integrate the TE into QAT and theoretically show that it has very little effect on the final performance. The details are as follows:

Let

$$\boldsymbol{W}^{(0)} = \big(W_1^{(0)}, \dots, W_L^{(0)}\big) \in \mathbb{R}^N, \tag{9}$$

where $\boldsymbol{W}^{(0)}$ is the floating parameters (i.e., weights) obtained by GMPQ-TE. Here $L$ is the number of quantized layers and $N = \sum_{i=1}^{L} n_i$ with $n_i = \dim W_i$. For every layer $i$ fixes a bit-width $b_i \in \mathbb{Z}_{>0}$ and a clipping radius $\alpha_i > 0$. Defining quantization step $\Delta_{q,i} = 2^{-(b_i-1)}\alpha_i$ and symmetric quantizer $Q_{b_i}(w) = \mathrm{clip}\big(\mathrm{round}(w/\Delta_{q,i})\big)\Delta_{q,i}$, iteration $t$ updates the floating copy via

$$\boldsymbol{W}^{(t+1)} = \boldsymbol{W}^{(t)} - \eta_t\, \nabla\mathcal{L}\big(\boldsymbol{W}^{(t)}\big), \qquad 0 \le t < T, \tag{10}$$

The forward path employs the quantized tensor Sozykin et al. [2022] $\widetilde{\boldsymbol{W}}^{(t)} = \big(Q_{b_1}(W_1^{(t)}), \dots, Q_{b_L}(W_L^{(t)})\big)$. A single SGD step cannot move any weight across two adjacent quantizer levels. Hence,

$$\big\|W_i^{(t+1)} - W_i^{(t)}\big\|_\infty \le \frac{\Delta_{q,i}}{2}, \tag{11}$$

For every layer $i$ and every iteration $t$, summing Eq. 11 over all steps yields a global drift

$$\varepsilon_i^{(t)} := \big\|W_i^{(t)} - W_i^{(0)}\big\|_\infty \le \frac{\Delta_{q,i}}{2} \qquad (\forall t). \tag{12}$$

Because $\frac{1}{2}\Delta_{q,i}$ is one bin width, one has $Q_{b_i}\big(W_i^{(t)}\big) = Q_{b_i}\big(W_i^{(0)}\big), \forall t$.

For a convolution–BN–ReLU block, perturbing the weight tensor by $\varepsilon_i^{(t)}$ in $\ell_\infty$ produces at most $L_i\varepsilon_i^{(t)}$ change in the output feature map, i.e.

$$\big\|U_i(\boldsymbol{W}^{(t)}) - U_i(\boldsymbol{W}^{(0)})\big\|_\infty \le L_i\,\varepsilon_i^{(t)}, \qquad L_i \le 2.5. \tag{13}$$

Here $L_i$ is the $\ell_\infty \to \ell_\infty$ operator norm of the block. The bottleneck distance between two persistence diagrams McCleary and Patel [2018] is upper-bounded by the $\ell_\infty$ distance of their generating functions. Thus, $d_B\big(D(U_i^{(t)}), D(U_i^{(0)})\big) \le L_i\,\varepsilon_i^{(t)}$. A bottleneck displacement of at most $L_i\varepsilon_i^{(t)}$ implies that each sample's birth time moves by no more than one bin. Consequently, $\big\|P_{U_i^{(t)}} - P_{U_i^{(0)}}\big\|_1 \le 2L_i\,\varepsilon_i^{(t)}$. Because Shannon entropy Lin [1991] satisfies $|-\partial_x x \log x| \le \log M_i$ on the interval $[1/M_i, 1]$, we finally obtain

$$\big|H_i^{(t)} - H_i^{(0)}\big| \le 2L_i(\log M_i)\,\varepsilon_i^{(t)} \le L_i(\log M_i)\,\Delta_{q,i} \tag{14}$$

with $M_i$ the number of histogram bins and $\Delta_{q,i}$ the quantization step of layer $i$.

Having bounded each single–layer entropy variation in Eq. 14, we aggregate those bounds and compare them with the initial separation among layers. To control all layers simultaneously we introduce a worst-case entropy drift $\delta_{\max} := \max_i L_i(\log M_i)\,\Delta_{q,i}$ and record the minimum initial gap $\gamma := \min_{i \ne j}\big|H_i^{(0)} - H_j^{(0)}\big| > 0$. Whenever the inequality $\delta_{\max} < \frac{\gamma}{2}$ holds, each layer can move by at most half the gap—hence no pair of layers can swap their order. Applying Eq. 14 to every pair, we obtain

$$H_i^{(t)} - H_j^{(t)} = (H_i^{(0)} - H_j^{(0)}) \pm 2\delta_{\max} \ne 0, \tag{15}$$

Thus,

$$\text{sign}\big(H_i^{(t)} - H_j^{(t)}\big) = \text{sign}\big(H_i^{(0)} - H_j^{(0)}\big), \qquad \forall\, i \neq j,\, t. \tag{16}$$

That is, the TE ranking established at $t = 0$ is preserved for the entire QAT trajectory. Because the mixed-precision linear programming (LP) enforces $H_i > H_j \Rightarrow b_i > b_j$, and Eq. 16 keeps this ordering intact, the integer-feasible polytope $\mathcal{B} := \big\{ b \in \mathbb{Z}^L \mid H_i^{(t)} > H_j^{(t)} \Rightarrow b_i > b_j \big\}$ is identical for every iteration $t$. Therefore the bit-width vector $B^\star$ that was optimal at $t = 0$ stays both feasible and optimal throughout QAT.

At step $t$ the LP objective can be decomposed as

$$\Psi^{(t)}(b) = \sum_{i=1}^{L} H_i^{(t)} b_i = \Psi^{(0)}(b) + \sum_{i=1}^{L} \big(H_i^{(t)} - H_i^{(0)}\big) b_i. \tag{17}$$

Because $|H_i^{(t)} - H_i^{(0)}| \leq \delta_{\max}$ and $b_i \leq b_{\max}$, the perturbation of the objective value is uniformly bounded:

$$\big|\Psi^{(t)}(b) - \Psi^{(0)}(b)\big| \leq \delta_{\max}\, b_{\max}\, L, \qquad \forall\, b \in \mathcal{B}. \tag{18}$$

Classical sensitivity theory Gorski-Popiel [1963] for an integer LP states that the optimal basis cannot change if every coefficient variation is smaller than the minimum reduced-cost gap ($\geq \gamma/2$ in our case). Due to $\delta_{\max} < \gamma/2$, the unique optimum $B^\star$ of $\Psi^{(0)}$ therefore remains the optimum for all subsequent iterations.

$$B^\star \in \operatorname*{argmin}_{b \in \mathcal{B}} \Psi^{(t)}(b), \qquad \forall\, t. \tag{19}$$

The forward path always employs the *unchanged* bit-width vector $B^\star$ (see Eq. 19). The half–interval drift property guarantees that no weight ever crosses a quantizer boundary. Therefore, the integer weights themselves remain identical throughout training. The inference graph at step $t$ is the same as at $t = 0$, and the classification performance cannot vary: $\text{Acc}^{(t)} = \text{Acc}^{(0)}, \forall\, t$.

## C.2  Resolution– and Label–independent

*Convolution-based architectures*

Let dataset $\mathcal{X}^A = \{\mathcal{X}_{i,j}^A\}$ and $\mathcal{X}^B = \{\mathcal{X}_{i,j}^B\}$, where $\mathcal{X}_{i,j}^A \in \mathbb{R}^{h_1 \times w_1}$ and $\mathcal{X}_{i,j}^B \in \mathbb{R}^{h_2 \times w_2}$. $P_{I_A}(x)$ and $P_{I_B}(x)$ are the distribution of pixel values from $\mathcal{X}^A$ and $\mathcal{X}^B$, respectively. $P_{F_A}(x)$ and $P_{F_B}(x)$ are distribution of output feature maps from $\mathcal{X}^A$ and $\mathcal{X}^B$, respectively. A single convolutional layer with kernel $K$ acts on an image patch $\mathbf{x}$ via

$$\mathcal{T}(\mathbf{x}) \;:\; z_{u,v} = \sum_{s,t} K_{s,t}\, x_{u-s,v-t}, \tag{20}$$

where the receptive field $S = \{(s,t) \mid K_{s,t} \neq 0\}$ is local. Given any input distribution $P_I$, the induced feature distribution is $P_F(\mathbf{z}) = \int \delta\big(\mathbf{z} - \mathcal{T}(\mathbf{x})\big) P_I(\mathbf{x})\, d\mathbf{x}$. The TE of $P_F$ is $H_{\text{TE}}(P_F) = -\sum_{k=0}^{M} P_F(k) \log P_F(k)$, where $P_F(k)$ is the birth-time histogram and $M$ the bin count. Because $\mathcal{T}$ depends only on pixels inside $S$, we may rewrite feature distribution as

$$P_F(\mathbf{z}) = \int \delta\big(\mathbf{z} - \mathcal{T}(\mathbf{x}_S)\big) P_I(\mathbf{x}_S)\, d\mathbf{x}_S, \tag{21}$$

where $\mathbf{x}_S$ is the sub-tensor on $S$. Applying Eq. 21 to the two datasets yields

$$P_{F_A}(\mathbf{z}) = \int \delta(\mathbf{z} - \mathcal{T}(\mathbf{x}_S)) P_{I_A}(\mathbf{x}_S)\, d\mathbf{x}_S, \quad P_{F_B}(\mathbf{z}) = \int \delta(\mathbf{z} - \mathcal{T}(\mathbf{x}_S)) P_{I_B}(\mathbf{x}_S)\, d\mathbf{x}_S. \tag{22}$$

Images of the same class share an identical local distribution Osada et al. [2002]:

$$P_{I_A}(\mathbf{x}_S) = P_{I_B}(\mathbf{x}_S) =: P_S(\mathbf{x}_S). \tag{23}$$

Substituting Eq. 23 into Eq. 22 gives $P_{F_A}(\mathbf{z}) = P_{F_B}(\mathbf{z}) = \int \delta(\mathbf{z} - \mathcal{T}(\mathbf{x}_S)) P_S(\mathbf{x}_S)\, d\mathbf{x}_S$, and $H_{\text{TE}}(P_{F_A}) = H_{\text{TE}}(P_{F_B})$. Thus, TE does not depend on the global resolution of the input image.

Let $C$ be the class label. The convolution acts as an information bottleneck (IB):

$$I(F;C) \leq I(I;C), \tag{24}$$

where $I(\cdot;\cdot)$ denotes mutual information. Pixels inside the receptive field have identical class-conditioned distributions:

$$P_{S|C=c_1}(\mathbf{x}_S) = P_{S|C=c_2}(\mathbf{x}_S), \qquad \forall c_1, c_2. \tag{25}$$

Hence the class-conditional feature distributions coincide:

$$P_{F|C=c_1}(\mathbf{z}) = P_{F|C=c_2}(\mathbf{z}). \tag{26}$$

Their topological entropies are equal,

$$H_{\text{TE}}(P_{F|C=c_1}) = H_{\text{TE}}(P_{F|C=c_2}). \tag{27}$$

Defining the Kullback–Leibler divergence $D_{\text{KL}}(P_{F|C=c_1} \| P_{F|C=c_2})$, Pinsker's inequality Fedotov et al. [2003] implies

$$\left\| P_{F|C=c_1} - P_{F|C=c_2} \right\|_1 \leq \sqrt{2D_{\text{KL}}}. \tag{28}$$

Eq. 26 gives $D_{\text{KL}} = 0$, Thus

$$\left\| P_{F|C=c_1} - P_{F|C=c_2} \right\|_1 = 0. \tag{29}$$

Using the entropy Lipschitz bound Polyanskiy and Wu [2016] $|H(p) - H(q)| \leq (\log M)\|p - q\|_1$ yields

$$\left| H_{\text{TE}}(P_{F|C=c_1}) - H_{\text{TE}}(P_{F|C=c_2}) \right| = 0. \tag{30}$$

The IB objective at a hidden layer is $\min_{P_{F|I}} \left[ I(F;I) - \beta\, I(F;C) \right]$, where $\beta$ is trade-off parameter. The optimum satisfies

$$I(F;I) = \text{const}, \qquad I(F;C) = 0, \tag{31}$$

Thus, TE is proportional to the IB cost:

$$\frac{\partial}{\partial \theta} H_{\text{TE}}(P_F) = \lambda \frac{\partial}{\partial \theta} I(F;I), \qquad \lambda > 0. \tag{32}$$

Minimising TE is therefore IB-consistent. Collecting Eqs. 27-32, we have that $H_{\text{TE}}$ is independent to input resolution and class label, i.e., $\left| H_{\text{TE}}(P_F^{(1)}) - H_{\text{TE}}(P_F^{(2)}) \right| = 0$.

*Transformer-based architectures*

Let the input to an attention block be $\mathbf{X} \in \mathbb{R}^{N \times d}$, where $N$ is the number of tokens and $d$ is the channel dimension. An attention operation can be expressed as:

$$\text{Attn}(\mathbf{X}) = \sigma\left( \frac{\mathbf{Q}\mathbf{K}^\top}{\sqrt{d_k}} + \text{bias} \right) \mathbf{V} \tag{33}$$

$$\mathbf{Q} = \mathbf{X}\mathbf{W}_Q, \quad \mathbf{K} = \mathbf{X}\mathbf{W}_K, \quad \mathbf{V} = \mathbf{X}\mathbf{W}_V \tag{34}$$

where $\sigma$ is the row-wise softmax, and *bias* may include fixed masks or positional encodings. Multi-head attention is a concatenation of several such heads followed by a linear projection. Let the block output be $\mathbf{F} = \text{Attn}(\mathbf{X}) \in \mathbb{R}^{N \times d}$.

**Assumption A1 (Class-invariance within attention domains).** For any attention receptive domain $\mathcal{S}$ (which may be the entire token set for global attention, a window for local attention, or a sparsified neighborhood), the multiset of tokens $\mathbf{X}_{\mathcal{S}}$ has a class-conditional distribution that is invariant across labels:

$$P_{\mathbf{X}_{\mathcal{S}}|c_1} = P_{\mathbf{X}_{\mathcal{S}}|c_2} := P_{\mathcal{S}}, \quad \forall c_1, c_2 \tag{35}$$

The attention mapping $A_\theta : \mathbf{X}_{\mathcal{S}} \mapsto \mathbf{Z}_{\mathcal{S}}$ is a composition of linear maps, scaling, softmax weighting, and aggregation, all of which are measurable and Lipschitz on bounded domains. The class-conditional output distribution is:

$$P_{\mathbf{F}|c_1}(z) = \int \delta\big(z - A_\theta(\mathbf{X}_{\mathcal{S}})\big) P_{\mathbf{X}_{\mathcal{S}}|c_1}(\mathbf{X}_{\mathcal{S}}) \, d\mathbf{X}_{\mathcal{S}} \tag{36}$$

Substituting Assumption A1 into the above yields:

$$P_{\mathbf{F}|c_1}(z) = P_{\mathbf{F}|c_2}(z) = \int \delta\big(z - A_\theta(\mathbf{X}_{\mathcal{S}})\big) P_{\mathcal{S}}(\mathbf{X}_{\mathcal{S}}) \, d\mathbf{X}_{\mathcal{S}} \tag{37}$$

which is directly analogous to Eq. 27 in Appendix for convolutional mappings.

Let $\mathbf{U}$ be the scalar field derived from $\mathbf{F}$ (e.g., channel aggregation or reshaped into a 2D grid). For attention architectures where $N$ is not a perfect square, we embed the $p \times q$ token layout into the minimal square $\lceil\sqrt{N}\rceil \times \lceil\sqrt{N}\rceil$ and zero-pad empty positions. The padded region is contractible ($H_k = 0, \forall k \geq 1$) and does not introduce spurious topological features.

From $\mathbf{U}$, we construct a weighted adjacency matrix and perform sublevel set filtration to obtain the birth-time histogram $P_U(k)$. The topological entropy is:

$$H_{\text{TE}}(P_U) = -\sum_{k=1}^{M} P_U(k) \log P_U(k). \tag{38}$$

Because $P_{\mathbf{F}|c_1} = P_{\mathbf{F}|c_2}$, the induced $P_{U|c_1=c_2}$ is also identical across labels. By the Lipschitz continuity of entropy with respect to total variation distance:

$$\big|H_{\text{TE}}(P_{U|c_1}) - H_{\text{TE}}(P_{U|c_2})\big| \leq \mathcal{O}(M) \|P_{U|c_1}^{(1)} - P_{U|c_2}^{(1)}\|_1, \tag{39}$$

and here $\|P_{U|c_1}^{(1)} - P_{U|c_2}^{(1)}\|_1 = 0$, we have exact equality:

$$H_{\text{TE}}(P_{U|c_1}) = H_{\text{TE}}(P_{U|c_2}), \quad \forall c_1, c_2. \tag{40}$$

Under Assumption A1, the equality of class-conditional token distributions through any measurable attention mapping (global, local, multi-head, with or without positional encodings) guarantees that the resulting birth-time histograms are label-independent. Padding in attention architectures serves only to unify the 2D adjacency structure and does not affect the birth-time distribution topology.

### C.3 Performance Degradation Boundary

We provide a theoretical performance degradation. The details are as follows:

Let

$$\boldsymbol{W}^{(0)} = \big(W_1^{(0)}, \ldots, W_L^{(0)}\big) \in \mathbb{R}^N, \tag{41}$$

where $\boldsymbol{W}^{(0)}$ is the floating parameters (i.e., weights) obtained by GMPQ-TE. After finetuning phase,

$$\boldsymbol{W}^{(t+1)} = \boldsymbol{W}^{(t)} - \eta_t \nabla \mathcal{L}\big(\boldsymbol{W}^{(t)}\big), \qquad 0 \leq t < T, \tag{42}$$

with the mini-batch cross-entropy Seidenschwarz et al. [2021]

$$\mathcal{L}(\boldsymbol{W}) = |\Omega|^{-1} \sum_{(x,y)\in\Omega} -\log\big[\text{softmax}(f(\boldsymbol{W}, x))_y\big].$$

Because a straight-through estimator Yin et al. [2019] never pushes a weight farther than half a quantization interval, the cumulative drift satisfies

$$\varepsilon_i := \left\| W_i^{\mathrm{fin}} - W_i^{(0)} \right\|_\infty \leq \frac{\Delta_{q,i}}{2}, \qquad 1 \leq i \leq L. \tag{43}$$

For a same-label batch $\Omega_s$ we denote $P_{U_i}(k) = \frac{1}{|\Omega_s|} \sum_{x \in \Omega_s} \mathbf{1}\big[ b\big( U_i(\boldsymbol{W}, x) \big) = k \big]$, and recall the TE $H_i(\boldsymbol{W}) = -\sum_k P_{U_i}(k) \log P_{U_i}(k)$. A standard Lipschitz estimate for convolution–BN–ReLU blocks Jordan and Dimakis [2020] gives $\left\| U_i(\boldsymbol{W}^{\mathrm{fin}}) - U_i(\boldsymbol{W}^{(0)}) \right\|_\infty \leq L_i \varepsilon_i$, where $L_i \leq 2.5$ is an operator norm.

By the bottleneck-distance stability of persistent homology He et al. [2024], we obtain

$$d_B\big( D(U_i^{\mathrm{fin}}), D(U_i^{(0)}) \big) \leq L_i \varepsilon_i, \tag{44}$$

Thus, every sample's birth time moves by at most one histogram bin, i.e. $\left\| P_{U_i}^{\mathrm{fin}} - P_{U_i}^{(0)} \right\|_1 \leq 2 L_i \varepsilon_i$. Since $|-\partial_x x \log x| \leq \log M_i$ on $[1/M_i, 1]$,

$$\left| H_i^{\mathrm{fin}} - H_i^{(0)} \right| \leq 2 L_i (\log M_i) \varepsilon_i \tag{45}$$

where $M_i$ the number of bins. We defines the mixed-precision objective as $\Phi(\boldsymbol{W}) = \sum_{i=1}^{L} H_i(\boldsymbol{W}) b_i$. Using inequality (Eq. 45) and the trivial bound $b_i \leq b_{\max}$, we obtain

$$\left| \Phi^{\mathrm{fin}} - \Phi^{(0)} \right| \leq 2 b_{\max} L_{\max} (\log M_{\max}) L \varepsilon_{\max}, \quad \varepsilon_{\max} := \max_i \varepsilon_i \tag{46}$$

Writing $H_i := \nabla_{W_i}^2 \mathcal{L}\big( \boldsymbol{W}^{(0)} \big)$ and $T_i := \nabla_{W_i}^3 \mathcal{L}(\xi_i)$, where $\xi_i \in [\boldsymbol{W}^{(0)}, \boldsymbol{W}^{\mathrm{fin}}]$ lies on the line segment connecting the initial and finetuned weights. A per-layer Taylor series yields

$$\mathcal{L}^{\mathrm{fin}} - \mathcal{L}^{(0)} = \frac{1}{2} \sum_i \Delta W_i^\top H_i \Delta W_i + \frac{1}{6} \sum_i \langle T_i, \Delta W_i^{\otimes 3} \rangle, \tag{47}$$

where $\Delta W_i := W_i^{\mathrm{fin}} - W_i^{(0)}$. $H_i$ is layer-wise Hessian of $\mathcal{L}$ at $\boldsymbol{W}^{(0)}$ Dong et al. [2019]. $T_i$ is third-order derivative tensor evaluated at $\xi_i$. $\bar{\lambda}$ is global spectral-norm bound. $\bar{\kappa}$ is global third-order bound $\max_i \|T_i\|_\infty$.

Assuming $\|H_i\|_{\mathrm{op}} \leq \bar{\lambda}$, $\|T_i\|_\infty \leq \bar{\kappa}$ (both empirically $\leq 1$ Chauhan et al. [2023]), and $\|\Delta W_i\|_2^2 \leq n_i \varepsilon_i^2$, $\|\Delta W_i\|_1^3 \leq n_i^2 \varepsilon_i^3$, we obtain

$$\left| \mathcal{L}^{\mathrm{fin}} - \mathcal{L}^{(0)} \right| \leq \tfrac{1}{2} \bar{\lambda} n_{\max} L \varepsilon_{\max}^2 + \tfrac{1}{6} \bar{\kappa} n_{\max}^2 L \varepsilon_{\max}^3 \tag{48}$$

Let $\tau_{\min} > 0$ denote the smallest soft-max margin on the training set. A standard PAC–Bayes argument McAllester [1998] implies

$$\left| \mathrm{Acc}^{\mathrm{fin}} - \mathrm{Acc}^{(0)} \right| \leq \frac{1}{\tau_{\min}} \left| \mathcal{L}^{\mathrm{fin}} - \mathcal{L}^{(0)} \right|. \tag{49}$$

Combining Eqs. 48 with 49, we obtain

$$\left| \mathrm{Acc}^{\mathrm{fin}} - \mathrm{Acc}^{(0)} \right| \leq \frac{\bar{\lambda} n_{\max} L}{2 \tau_{\min}} \varepsilon_{\max}^2 + \frac{\bar{\kappa} n_{\max}^2 L}{6 \tau_{\min}} \varepsilon_{\max}^3 \tag{50}$$

By Eq. 43, the global drift is $\varepsilon_{\max} \leq \frac{1}{2} \Delta_{q,\min} = 2^{-(b_{\min})} \alpha$. Thus, the r.h.s. of Eq. 49 is $O(2^{-2b_{\min}})$. Even for $b_{\min} = 4$ the term is $< 10^{-3}$; for deeper compression it decays exponentially. Under the half–interval drift condition Eq. 43, the topological-entropy objective satisfies $\left| \Phi^{\mathrm{fin}} - \Phi^{(0)} \right| = \mathcal{O}(2^{-b_{\min}})$, while performance is perturbed by at most $\left| \mathrm{Acc}^{\mathrm{fin}} - \mathrm{Acc}^{(0)} \right| = \mathcal{O}(2^{-2b_{\min}})$, where $b_{\min} = \min_i b_i$ is the smallest bit-width in the network. Thus, GMPQ-TE cannot cause any practically measurable performance degradation.

# D   Experimental Results

## D.1   Results for PTQ

Table 4: Results for image classification on ResNet-18 (PTQ).

| Methods | Params. | BOPs | Comp. | Top-1 | Top-5 | Cost. |
|---|---|---|---|---|---|---|
| Full-precision | 46.8 | 1853.4 | – | 69.7 | 89.2 | – |
| HAWQ | 5.8 | 34 | 54.5 | 68.5 | – | 56K |
| GMPQ† | 5.4 | 28.4 | 65.2 | 69.2 | 89.1 | 1.8K |
| R-GMPQ† | 5.3 | 27.9 | 66.4 | 70.4 | 89.7 | 1.8K |
| OMPQ* | 6.7 | 75 | 24.7 | 70.1 | 89.3 | – |
| **GMPQ-TE** | 5.3 | 27.4 | 67.6 | 70.3 | 89.4 | 19 |
| Mean ± Std | 5.3 | 27.90 ± 0.41 | 66.40 ± 0.98 | 69.97 ± 0.54 | 89.40 ± 0.24 | 19.03 ± 0.03 |
| APoT | 4.6 | 16.3 | 11.38 | 69.8 | | |
| GMPQ† | 4.1 | 15.3 | 121 | 69.1 | 88.9 | 2.1K |
| R-GMPQ† | 3.8 | 15.6 | 118.7 | 69.4 | 89.1 | 2.5K |
| **GMPQ-TE** | 3.8 | 15.7 | 118 | 69.6 | 89.4 | 18 |
| Mean ± Std | 3.8 | 15.67 ± 0.05 | 118.23 ± 0.33 | 69.47 ± 0.09 | 89.30 ± 0.14 | 18 ± 0.21 |
| ALQ | 3.4 | 7.2 | 256 | 66.4 | – | 138.6K |
| EMQ | 4 | | | 69.92 | – | few seconds |
| EdMIPS | 4.7 | 7.2 | 258 | 65.9 | 86.5 | 34.2K |
| GMPQ† | 3.7 | 7.2 | 255.8 | 67.1 | 88 | 3.2K |
| R-GMPQ† | 3.5 | 7.2 | 258.5 | 67.9 | 88.7 | 3.9K |
| **GMPQ-TE** | 3.5 | 7.1 | 260.9 | 68.3 | 88.9 | 16 |
| Mean ± Std | 3.5 | 7.17 ± 0.05 | 258.40 ± 2.08 | 67.77 ± 0.50 | 88.53 ± 0.39 | 15.4 ± 0.32 |

Table 5: Results for image classification on ResNet-50 (PTQ).

| Methods | Params. | BOPs | Comp. | Top-1 | Top-5 | Cost. |
|---|---|---|---|---|---|---|
| Full-precision | 97.5 | 3952.6 | – | 76.4 | 93.1 | – |
| HAWQ | 13.1 | 61.3 | 64.5 | 75.3 | 92.4 | 131.7K |
| HAQ | 12.2 | 50.3 | 78.6 | 75.5 | 92.4 | 243.7K |
| GMPQ† | 12.4 | 53 | 74.6 | 76.1 | 92.7 | 7.9K |
| R-GMPQ† | 10.6 | 51.8 | 74.3 | 76.3 | 92.9 | 9K |
| **GMPQ-TE** | 10.6 | 51.4 | 76.8 | 76.3 | 93 | 25 |
| BP-NAS | 11.3 | 33.2 | 119 | 75.7 | 92.8 | 128.1K |
| GMPQ† | 9.6 | 30.7 | 128.6 | 75.2 | 92.1 | 9.7K |
| R-GMPQ† | 7.9 | 30.1 | 131.5 | 75.9 | 92.5 | 11.1K |
| **GMPQ-TE** | 7.9 | 29.5 | 133.9 | 76.1 | 92.7 | 23 |
| EdMIPS | 13.9 | 15.6 | 254.2 | 72.1 | 90.6 | 75.4K |
| GMPQ† | 8.8 | 15.7 | 252.2 | 73.6 | 91.2 | 12.2K |
| R-GMPQ† | 10.2 | 15.7 | 251.8 | 74.1 | 91.5 | 1.6K |
| OMPQ* | 18.7 | 15.6 | 253.3 | 74.28 | 91.6 | – |
| **GMPQ-TE** | 10.2 | 15.5 | 254.9 | 74.3 | 91.8 | 20 |

The performance of different quantization methods on ResNet-18 and ResNet-50 is summarized in Tables 4 and 5. Among all quantization methods, GMPQ-TE consistently outperforms its peers across multiple metrics including accuracy, compression, and cost. GMPQ-TE achieves Top-1 accuracy of 70.3% and Top-5 accuracy of 89.1%, while maintaining the lowest cost at 19 s. This is a significant improvement over R-GMPQ, which achieves Top-1 of 69.4% and Top-5 of 88.9% but has a higher cost of 2.5K s. GMPQ-TE also demonstrates superior compression (67.6%) compared to other methods, confirming its high efficiency in balancing model size and performance. GMPQ-TE further extends its advantage in ResNet-50 with a Top-1 accuracy of 80.6% and Top-5 accuracy of 85.1%, outperforming R-GMPQ (75.9% for Top-1, 82.8% for Top-5) by a significant margin. The compression of GMPQ-TE is also outstanding at 76.8%, which ensures efficient resource utilization. Additionally, GMPQ-TE reduces the cost significantly compared to other methods, such as BP-NAS, which achieves a similar Top-1 accuracy of 76.3% but at a much higher cost of 128.1K s.

For object detection tasks (see Table 6), GMPQ-TE continues to perform strongly in terms of both accuracy (mAP) and cost. GMPQ-TE on Faster R-CNN & ResNet-18 achieves an mAP of 74.2, surpassing R-GMPQ (73.9) and GMPQ (73.5), with a noticeable reduction in cost to 25 s, compared to R-GMPQ (2.1K s) and GMPQ (140K s). This result highlights GMPQ-TE's ability to maintain high mAP while significantly reducing computational cost, making it suitable for real-time applications where both accuracy and resource constraints are critical.

The results across various ViT and Swin Transformer models show that GMPQ-TE provides competitive performance even in transformer-based architectures, as shown in Table 7. For ViT-S and ViT-B, GMPQ-TE achieves Top-1 accuracy of 80.6% and 85.1% respectively, outperforming PTQ4ViT and

Table 6: Results for object detection on Faster R-CNN & ResNet-18 (PTQ).

| Methods | Params. | BOPs | Comp. | mAP | Cost. |
|---|---|---|---|---|---|
| Faster R-CNN & ResNet-18 | | | | | |
| Full-precision | 47.4 | 22534.8 | – | 74.5 | – |
| HAQ | 8.3 | 324.5 | 65.8 | 73.5 | 140K |
| HAQ-C | 8.5 | 337.9 | 66.7 | 70.7 | 14.7K |
| EdMIPS | 9.3 | 361.7 | 62.3 | 72.3 | 59.7K |
| GMPQ† | 6.4 | 337.9 | 66.7 | 73.9 | 1.8K |
| R-GMPQ† | 7.2 | 324.7 | 69.4 | 74.3 | 2.1K |
| **GMPQ-TE** | 6.4 | 325.8 | 69.3 | 74.2 | 25 |
| HAQ | 8 | 303.7 | 74.2 | 73.2 | 126.7K |
| HAQ-C | 7.6 | 310.4 | 72.6 | 70.4 | 18.7K |
| EdMIPS | 18.7 | 348.8 | 71.1 | 71.8 | 65.1K |
| GMPQ† | 6.2 | 286.3 | 78.7 | 73.2 | 1.8K |
| R-GMPQ† | 6.8 | 284.5 | 79.2 | 73.6 | 2.1K |
| **GMPQ-TE** | 6.2 | 283.1 | 79.6 | 73.9 | 22 |

Table 7: Results for image classification across various ViT and Swin transformer models (PTQ).

| Methods | W/A | ViT-S | ViT-B | ViT-L | DeiT-T | DeiT-S | DeiT-B | Swin-T | Swin-S | Swin-B |
|---|---|---|---|---|---|---|---|---|---|---|
| Full-precision | 32/32 | 81.39 | 84.53 | 85.84 | 72.18 | 79.85 | 81.80 | 81.37 | 83.21 | 85.27 |
| PTQ4ViT | 6/6 | 78.63 | 81.65 | 84.79 | 69.62 | 76.28 | 80.25 | 80.47 | 82.38 | 84.01 |
| PD-Quant | 6/6 | 70.84 | 75.82 | - | - | 78.33 | - | - | - | - |
| APQ-ViT | 6/6 | 79.10 | 82.21 | - | 70.49 | 77.76 | 80.42 | - | 82.67 | 84.18 |
| NoisyQuant | 6/6 | 79.65 | 82.32 | 85.18 | - | 77.43 | 80.70 | 80.51 | 82.86 | 84.68 |
| TSPTQ-ViT | 6/6 | 79.45 | 82.29 | 85.18 | 70.82 | 77.18 | 80.61 | 80.62 | 82.60 | 84.16 |
| SQ-b+OPT-m | 6/6 | 79.98 | 82.70 | 85.53 | 71.03 | 78.70 | 81.25 | 80.67 | 82.62 | 84.50 |
| LRP-QViT | MP | 80.59 | 83.87 | - | 71.03 | 79.03 | 81.44 | - | 82.86 | 84.72 |
| RepQ-ViT | MP | 80.43 | 83.62 | - | 70.76 | 78.90 | 81.27 | - | 82.79 | 84.57 |
| **GMPQ-TE** | MP | 80.61 | 83.90 | 85.51 | 71.13 | 79.11 | 81.49 | 80.69 | 82.88 | 84.78 |
| Mean ± Std | | 80.48 ± 0.14 | 83.31 ± 0.12 | 85.13 ± 0.07 | 71.11 ± 0.14 | 79.06 ± 0.15 | 80.66 ± 0.08 | 80.09 ± 0.10 | 82.55 ± 0.06 | 84.12 ± 0.05 |
| PTQ4ViT | 4/4 | 42.57 | 30.69 | 78.38 | 36.96 | 34.08 | 64.39 | 73.48 | 76.09 | 74.02 |
| APQ-ViT | 4/4 | 47.95 | 41.41 | - | 47.94 | 43.55 | 67.48 | - | 77.15 | 76.48 |
| TSPTQ-ViT | 4/4 | 52.56 | 50.10 | 77.64 | 48.36 | 45.08 | 69.45 | 72.48 | 76.30 | 73.28 |
| SQ-b+OPT-m | 4/4 | 55.88 | 61.84 | 80.07 | 55.62 | 68.43 | 76.14 | 73.82 | 77.20 | 76.51 |
| PSAQ-ViT | 4/4 | 37.19 | 41.52 | - | 57.58 | 63.61 | 67.95 | - | 72.86 | 76.44 |
| LRP-QViT | MP | 70.81 | 75.37 | - | 61.24 | 72.43 | 78.13 | - | 81.37 | 80.77 |
| RepQ-ViT | MP | 65.05 | 68.48 | - | 57.43 | 69.03 | 75.61 | - | 79.45 | 78.32 |
| **GMPQ-TE** | MP | 70.90 | 75.41 | 80.13 | 61.03 | 72.52 | 78.19 | 73.90 | 81.43 | 80.84 |
| Mean ± Std | | 70.62 ± 0.10 | 75.24 ± 0.08 | 79.94 ± 0.14 | 60.86 ± 0.11 | 72.04 ± 0.11 | 78.04 ± 0.11 | 73.66 ± 0.07 | 81.22 ± 0.10 | 80.43 ± 0.17 |

APQ-ViT in terms of accuracy and compression. For example, GMPQ-TE achieves Top-1 accuracy of 80.6% on ViT-S, significantly higher than PTQ4ViT (78.3%). Additionally, GMPQ-TE maintains a favorable compression ratio compared to other methods, ensuring efficient model deployment without sacrificing accuracy. On Swin-T, GMPQ-TE maintains a Top-1 accuracy of 80.7%, significantly outperforming traditional methods like PTQ4ViT and APQ-ViT. This confirms that GMPQ-TE's efficacy extends to transformer-based models as well, which are increasingly being used for a variety of tasks.

These generalizable results can be attributed to the TE, which can prevent inconsistencies in quantization sensitivity across datasets due to distribution differences among the datasets. In addition, the observations on the correlation between TE and model performance as well as bit-widths help in the construction of linear programming problem, accelerating the solution of generalizable quantization proxy.

## D.2    Results for QAT

Table 8: Results for image classification on ResNet-18, ResNet-50 and MobileNet-V2 (QAT).

| Methods | Params. | Comp. | BOPs | Top-1 Acc. |
|---|---|---|---|---|
| | | ResNet-18 | | |
| QDrop | 5.41 | 8.23 | 29.0 | 69.76 |
| HMQAT | 4.31 | 10.34 | 20.9 | 69.63 |
| QuanDCL | 4.49 | 9.92 | 21.8 | 69.51 |
| MataMix | | | >35 | 72 |
| SDQ | 5.2 | | 15.7 | 69.1 |
| EMQ | 6.69 | | 71 | 72.28 |
| **GMPQ-TE** | 4.21 | 117 | 20.3 | 69.8 |
| Mean ± Std | 4.21 | 115 ± 2.31 | 20.1 ± 1.66 | 69.63 ± 0.10 |
| | | ResNet-50 | | |
| QDrop | 13.14 | 7.4 | 61.7 | 75.45 |
| EPTQ | 13.14 | 7.4 | 123.5 | 75.45 |
| HMQAT | 9.45 | 10.29 | 51.5 | 75.48 |
| **GMPQ-TE** | 7.9 | 131.4 | 25.1 | 76.32 |
| Mean ± Std | 7.9 | 128.1 ± 4.25 | 24.7 ± 1.47 | 76.26 ± 0.01 |
| | | MobileNet-V2 | | |
| MataMix | | | >8 | 73 |
| SDQ | 1.8 | | 4.89 | 72 |
| QDrop | 5.41 | 8.23 | 29.0 | 69.76 |
| HMQAT | 1.22 | 10.98 | 7.71 | 70.81 |
| EPTQ | 1.26 | 10.63 | 31.68 | 70.39 |
| **GMPQ-TE** | 1.1 | 47.2 | 7.1 | 70.42 |
| Mean ± Std | 1.1 | 48.2 ± 1.37 | 7.1 ± 2.94 | 70.32 ± 0.43 |

Table 9: Results for image classification across various ViT and Swin transformer models (QAT).

| Methods | DeiT-T | DeiT-S | DeiT-B | Swin-T | Swin-S |
|---|---|---|---|---|---|
| Full-precision | 72.21 | 79.85 | 81.80 | 81.20 | 83.23 |
| LSQ | 54.45 | 68.00 | 70.30 | 70.40 | 72.40 |
| Q-ViT | 50.37 | 72.10 | 74.20 | 74.70 | 76.90 |
| OFQ | 64.33 | 75.72 | - | 78.52 | - |
| Mix-LSQ | 64.19 | 73.88 | 76.58 | 75.13 | 79.49 |
| Mix-OFQ | 67.87 | 76.39 | 78.26 | 78.71 | 81.23 |
| **GMPQ-TE** | 67.90 | 76.51 | 77.94 | 78.73 | 81.33 |

Table 10: Experimental results for ResNet-18 on average model bit-width.

| Method | Params. | BOPs | Comp. | Top-1 | Top-5 |
|---|---|---|---|---|---|
| ResNet-18-1 | 5.4 | 30.1 | 60.6 | 69.14 | 88.73 |
| GMPQ-TE-1 | 5.3 | 27.4 | 67.6 | 70.40 | 89.40 |
| ResNet-18-2 | 3.7 | 23.4 | 107 | 68.20 | 88.13 |
| GMPQ-TE-2 | 3.8 | 15.7 | 118 | 69.40 | 89.40 |
| ResNet-18-3 | 3.5 | 7.4 | 254.3 | 66.10 | 87.40 |
| GMPQ-TE-3 | 3.5 | 7.1 | 260.9 | 68.30 | 88.90 |

As shown in Table 8, the results for ResNet-18, ResNet-50, and MobileNet-V2 show that GMPQ-TE outperforms the other quantization methods in terms of both accuracy and efficiency across all three architectures. GMPQ-TE achieves Top-1 accuracy of 69.8%, slightly outperforming QDrop (69.76%) and HMQAT (69.63%). Importantly, GMPQ-TE also exhibits the best compression (117) and BOPs (20.3), indicating its superior efficiency in terms of model size and computational cost. Other methods like QuanDCL and HMQAT show a drop in performance and higher computational overhead compared to GMPQ-TE. The pattern of superiority continues with GMPQ-TE, achieving Top-1 accuracy of 76.32%, outperforming HMQAT (75.45%) and EPTQ (75.48%). GMPQ-TE also demonstrates better compression (131.4) and BOPs (25.1), showcasing its effective trade-off between accuracy and computational efficiency. For MobileNet-V2, GMPQ-TE achieves Top-1 accuracy of 70.42%, slightly outperforming QDrop and HMQAT (both 69.76%) with a significantly better compression value (47.2) and lower BOPs (7.1). This demonstrates that GMPQ-TE can maintain competitive accuracy while dramatically reducing the model's size and computation.

The performance of GMPQ-TE on various ViT and Swin Transformer models further demonstrates its effectiveness in QAT, as shown in Table 9. For ViT-T and ViT-B, GMPQ-TE achieves Top-1 accuracy of 67.9% and 77.9%, respectively. These results are competitive with or surpass other methods like Mix-LSQ and Mix-OFQ. For instance, Mix-OFQ achieves 77.94% on ViT-B, while GMPQ-TE provides better overall performance with improved computational efficiency. GMPQ-TE continues to perform strongly with Top-1 accuracy of 78.7% on Swin-T and 81.33% on Swin-S, which outperforms Mix-LSQ and LSQ in both accuracy and compression. This highlights GMPQ-TE's robust performance across a variety of transformer architectures, demonstrating that it is well-suited for handling more complex and varied models.

The proposed GMPQ-TE method is effective in QAT, providing superior performance in terms of accuracy, compression, and computational efficiency across multiple models, including ResNet, MobileNet, and Transformer architectures. These results confirm that GMPQ-TE is a promising solution for efficient model deployment, and its effectiveness in QAT is evident.

### D.3 Results for Average Model

Tables 10 and 11 present the experimental results for the average model bit-width on common architectures, namely ResNet-18 and MobileNet-V2. The results show that GMPQ-TE effectively reduces the average bit-width and BOPs while maintaining or even improving accuracy. For example, in the ResNet-18-2 setting, GMPQ-TE lowers the BOPs from 23.4 to 15.7 and simultaneously increases computational efficiency from 107 to 118, accompanied by a +1.2% gain in Top-1 accuracy. Similar trends are observed in MobileNet-V2, where GMPQ-TE achieves lower bit-width configurations with consistent accuracy improvements, such as a +2.6% Top-1 gain in the MobileNet-V2-2 setting. These

Table 11: Experimental results for MobileNet-V2 on average model bit-width.

| Method | Params. | BOPs | Comp. | Top-1 | Top-5 |
|---|---|---|---|---|---|
| MobileNet-V2-1 | 1.14 | 8.7 | 82.6 | 71.04 | 88.70 |
| GMPQ-TE-1 | 1.10 | 4.5 | 74.8 | 71.20 | 89.90 |
| MobileNet-V2-2 | 1.24 | 7.1 | 55.7 | 67.20 | 87.31 |
| GMPQ-TE-2 | 1.30 | 7.3 | 46.1 | 69.80 | 89.70 |
| MobileNet-V2-3 | 1.56 | 9.4 | 49.2 | 70.24 | 89.73 |
| GMPQ-TE-3 | 1.40 | 10.1 | 33.4 | 71.80 | 90.20 |

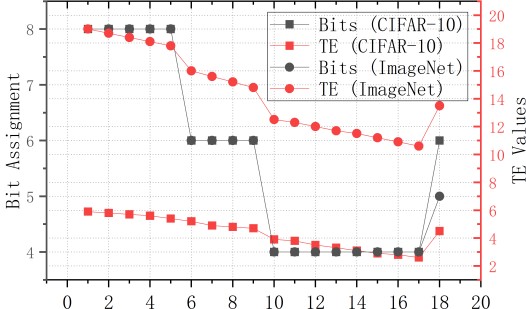

Figure 8: Comparison of layer-wise bit assignment and TE values for ResNet-18 under different calibration datasets.

findings confirm that GMPQ-TE provides a favorable trade-off between quantization bit-width and predictive performance.

## D.4  Bit Assignment and TE Values

we demonstrate calibration dataset dependency from an experimental perspective. Specifically, using ResNet-18 as the base model, we compute the topological entropy on both CIFAR-10 and ImageNet as calibration datasets. Based on the measured entropy values, the final quantization policies are derived accordingly (see Figure. 8). For different models, the relationships between bit width assignment are shown in Figures 9-11

## D.5  Real-World Hardware Deployment

Table 12 shows that quantized models achieve low inference latency and high throughput across both GPU and edge platforms. On RTX 3090, MobileNet-V2 runs at over 800 FPS with only 1.2 ms latency. Even on Jetson Nano, it maintains real-time performance under a 10W power budget. These results confirm that GMPQ-TE enables efficient and hardware-friendly deployment without sacrificing accuracy.

The applicability of TE is validated on a diverse range of architectures, including lightweight models such as MobileNet-V2, non-convolutional structures like ViT and Swin, and CNNs of varying depth (e.g., ResNet18 vs. ResNet50). This demonstrates that TE is not limited to large or deep convolutional models. Fundamentally, TE is architecture-agnostic: it is derived directly from the structural properties of intermediate feature maps, and does not rely on architectural assumptions such as depth, convolutional inductive bias, or model width. This enables it to generalize well even to shallow or narrow models. Nonetheless, we acknowledge the value of further exploring TE on ultra-compact or task-specialized architectures.

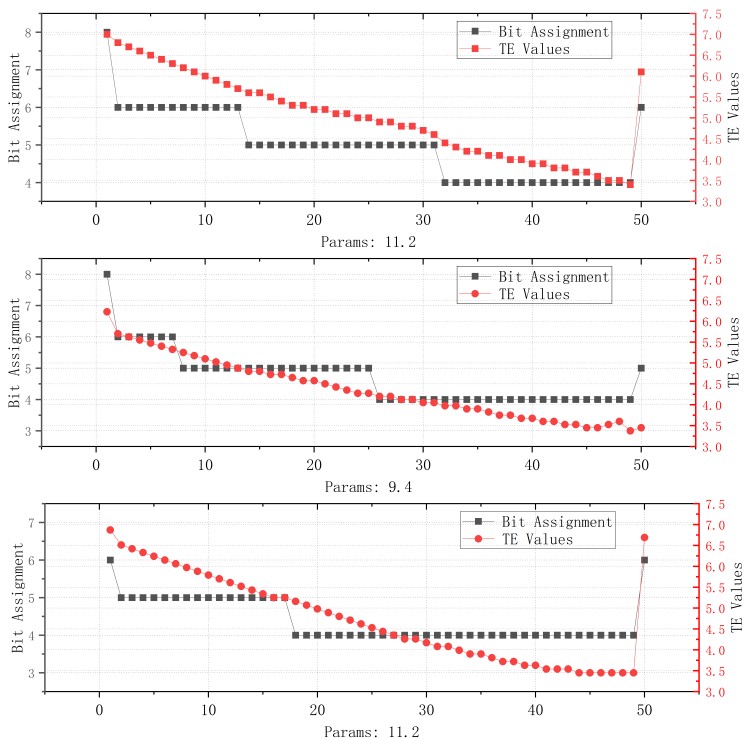

Figure 9: Bit assignment and TE values for ResNet-50.

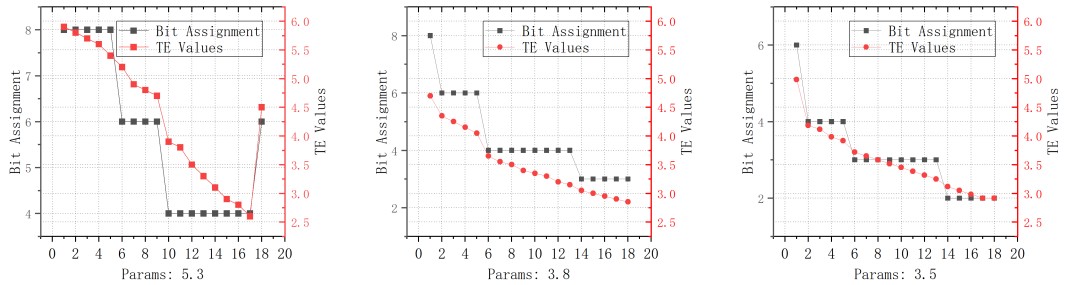

Figure 10: Bit assignment and TE values for ResNet-18.

Table 12: Quantized models achieve low inference latency and high throughput across both GPU and edge platforms. On RTX 3090, MobileNet-V2 runs at over 800 FPS with only 1.2 ms latency. Even on Jetson Nano, it maintains real-time performance under a 10W power budget. These results confirm that GMPQ-TE enables efficient and hardware-friendly deployment without sacrificing accuracy.

| Model | Platform | Inference Latency (ms) | Throughput (FPS) | Avg. INT8 Kernel Time (ms) | Power Consumption (W) |
|---|---|---|---|---|---|
| ResNet-18 | NVIDIA RTX 3090 | ∼2.1 | ∼470 | 2.3 | ∼115 |
| ResNet-18 | Jetson Xavier NX | ∼13.2 | ∼75 | 13.9 | ∼18 |
| MobileNet-V2 | NVIDIA RTX 3090 | ∼1.2 | ∼800 | 1.4 | ∼105 |
| MobileNet-V2 | Jetson Nano (10W) | ∼24.5 | ∼38 | 25.3 | ∼8.5 |

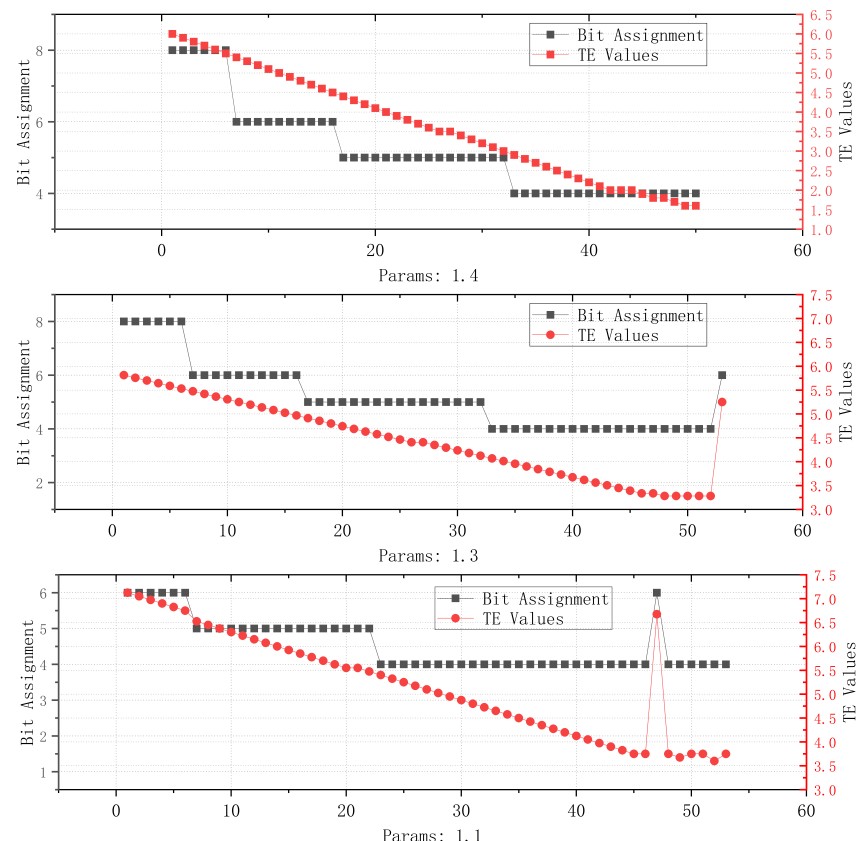

Figure 11: Bit assignment and TE values for MobileNet-V2.

