# OpenReview forum: "Efficient and Generalizable Mixed-Precision Quantization via Topological Entropy"
_NeurIPS.cc/2025/Conference — NeurIPS 2025 poster_

### Official Review · Reviewer_E1Lz · 2025-06-28

**Clarity:** 3
**Significance:** 2
**Originality:** 2
**Rating:** 4
**Confidence:** 3

**Summary:**

The paper introduces a mixed-precision quantization method (GMPQ-TE) based on topological entropy for compressing deep neural networks to fit resource-constrained devices. Using topological data analysis, the method measures layer-wise quantization sensitivity with topological entropy, enabling a single-pass linear programming approach to determine the quantization policy. Experiments demonstrate that a policy trained on CIFAR-10 generalizes to ImageNet and PASCAL VOC.

**Questions:**

Besides the weaknesses above:

The results in Tables 4, 7, and 8 lack error bars or confidence intervals. Please provide a statistical analysis and specify the method used for computation.

Hardware deployment is mentioned but lacks specific performance metrics. Please provide detailed deployment results for platforms such as GPUs, TPUs, or edge devices.

The paper does not sufficiently compare topological entropy to other quantization sensitivity metrics. Please include comparative experiments or theoretical analysis to highlight its unique advantages.

**Ethical Concerns:**

["NO or VERY MINOR ethics concerns only"]

**Final Justification:**

Thank the authors for the submission and the feedback during the rebuttal. Also, thanks for the other reviewers' comments, which all help the reviewer to finalize the evaluation (still keep positive).

**Limitations:**

yes

**Quality:**

3

**Strengths And Weaknesses:**

Strengths

The application of topological entropy to mixed-precision quantization is a novel contribution, leveraging topological data analysis to assess quantization sensitivity. By analyzing the topological structures of feature maps via Betti curves and persistent homology, the method links quantization sensitivity to topological entropy.

The method’s efficiency and generalization are impactful for deploying models on resource-constrained devices (e.g., mobile platforms). Its cross-dataset generalization reduces the need for dataset-specific policy searches, and the label-independence of topological entropy enhances its applicability across diverse scenarios.

GMPQ-TE employs a single-pass linear programming approach to assign bit-widths based on topological entropy, thereby avoiding the iterative search overhead associated with traditional methods.

Weaknesses:

Experimental results (Tables 4, 7, 8) lack statistical significance analysis, which may undermine confidence in their reliability. The evaluation is limited to vision tasks with no exploration of non-vision tasks or non-convolutional architectures.

The paper does not sufficiently compare topological entropy to other quantization sensitivity metrics, limiting the demonstration of its unique advantages.

Real-world hardware deployment is mentioned, but it lacks detailed performance metrics, which restricts the evaluation of its practical applicability. The applicability of topological entropy to small networks or specific architectures remains underexplored.

---

> ### Author Rebuttal · Authors · 2025-07-30
>
> **We sincerely thank reviewer for pointing out these important issues**
>
> **Ans. For W1:**
>
> To address concern on statistical significance, we provide standard deviation and mean statistics for GMPQ-TE in Tables 4, 7, and 8. The “Mean ± Std” rows offer evidence of performance stability across diverse models. In terms of model diversity, our experiments include non-convolutional architectures such as Vision Transformers (ViT and Swin), which differ fundamentally from CNNs in design. While current evaluation focuses on vision tasks, we acknowledge limitation and consider extension to non-vision domains, including language and graph-based tasks, as an important future direction. We plan to explore these settings in follow-up work to further validate generality of our approach.
>
> *Table 4: Results for image classification on ResNet-18 (PTQ)*
> | Methods        | Params.   | BOPs     | Comp.     | Top-1     | Top-5     | Cost  |
> |----------------|-----------|----------|-----------|-----------|-----------|--------|
> | Full-precision | 46.8      | 1853.4   | –         | 69.7      | 89.2      | –      |
> | HAWQ           | 5.8       | 34       | 54.5      | 68.5      | –         | 56K    |
> | GMPQ†          | 5.4       | 28.4     | 65.2      | 69.2      | 89.1      | 1.8K   |
> | R-GMPQ†        | 5.3       | 27.9     | 66.4      | 70.4      | 89.7      | 1.8K   |
> | OMPQ∗          | 6.7       | 75       | 24.7      | 70.1      | 89.3      | –      |
> | **GMPQ-TE**    | 5.3       | 27.4     | 67.6      | 70.3      | 89.4      | 19     |
> | → Mean ± Std   | 5.3 | 27.90±0.41 | 66.40±0.98 | 69.97±0.54 | 89.40±0.24 | 19.03±0.03 |
> | APoT           | 4.6       | 16.3     | 11.38     | 69.8      | –         | –      |
> | GMPQ†          | 3.8       | 15.7     | 118       | 69.4      | 89.4      | 2.1K   |
> | R-GMPQ†        | 3.8       | 15.6     | 118.7     | 69.4      | 89.1      | 2.5K   |
> | **GMPQ-TE**    | 3.8       | 15.7     | 118       | 69.6      | 89.4      | 18     |
> | → Mean ± Std   | 3.8 | 15.67±0.05 | 118.23±0.33 | 69.47±0.09 | 89.30±0.14 | 18±0.21|
> | ALQ            | 3.4       | 7.2      | 256       | 66.4      | –         | 138.6K |
> | EdMIPS         | 4.7       | 7.2      | 258       | 65.9      | 86.5      | 34.2K  |
> | GMPQ†          | 3.7       | 7.2      | 255.8     | 67.1      | 88        | 3.2K   |
> | R-GMPQ†        | 3.5       | 7.2      | 258.5     | 67.9      | 88.7      | 3.9K   |
> | **GMPQ-TE**    | 3.5       | 7.1      | 260.9     | 68.3      | 88.9      | 16     |
> | → Mean ± Std   | 3.5| 7.17±0.05 | 258.40±2.08 | 67.77±0.50 | 88.53±0.39 | 15.4±0.32 |
>
> *Table 7: Results for image classification across various ViT and Swin transformer models (PTQ).*
> | Methods       | W/A   | ViT-S     | ViT-B     | ViT-L     | DeiT-T    | DeiT-S    | DeiT-B    | Swin-T    | Swin-S    | Swin-B    |
> |---------------|-------|-----------|-----------|----|----|---|-----|------|------|-----------|
> | Full-precision| 32/32 | 81.39     | 84.53     | 85.84     | 72.18     | 79.85     | 81.80     | 81.37     | 83.21     | 85.27     |
> | PTQ4ViT       | 6/6   | 78.63     | 81.65     | 84.79     | 69.62     | 76.28     | 80.25     | 80.47     | 82.38     | 84.01     |
> | PD-Quant      | 6/6   | 70.84     | 75.82     | -         | -         | 78.33     | -         | -         | -         | -         |
> | APQ-ViT       | 6/6   | 79.10     | 82.21     | -         | 70.49     | 77.76     | 80.42     | -         | 82.67     | 84.18     |
> | NoisyQuant    | 6/6   | 79.65     | 82.32     | -         | -         | 77.43     | 80.70     | 80.51     | 82.86     | 84.68     |
> | TSPTQ-ViT     | 6/6   | 79.45     | 82.29     | 85.18     | 70.82     | 77.18     | 80.61     | 80.62     | 82.60     | 84.16     |
> | SQ-b+OPT-m    | 6/6   | 79.98     | 82.70     | 85.53     | 71.03     | 78.70     | 81.25     | 80.67     | 82.62     | 84.50     |
> | LRP-QViT      | MP    | 80.59     | 83.87     | -         | 71.03     | 79.93     | 81.44     | -         | 82.86     | 84.72     |
> | RepQ-ViT      | MP    | 80.43     | 83.62     | -         | 70.76     | 78.90     | 81.27     | -         | 82.79     | 84.57     |
> | **GMPQ-TE**   | MP    | 80.61     | 83.90     | 85.51     | 71.13     | 79.11     | 81.49     | 80.69     | 82.88     | 84.78     |
> | → Mean ± Std  |       | 80.48±0.14| 83.31±0.12| 85.13±0.07| 71.11±0.14| 79.06±0.15| 80.66±0.08| 80.09±0.10| 82.55±0.06| 84.12±0.05 |
> | PTQ4ViT       | 4/4   | 42.57     | 30.69     | 78.38     | 36.96     | 34.08     | 64.39     | 73.48     | 76.09     | 74.02     |
> | APQ-ViT       | 4/4   | 47.95     | 41.41     | -         | 47.94     | 43.55     | 67.48     | -         | 77.15     | 76.48     |
> | TSPTQ-ViT     | 4/4   | 52.56     | 50.10     | 77.64     | 48.36     | 45.08     | 69.45     | 72.48     | 76.30     | 73.28     |
> | SQ-b+OPT-m    | 4/4   | 55.88     | 61.84     | 80.07     | 55.62     | 68.43     | 76.14     | 73.82     | 77.20     | 76.51     |
> | PSAQ-ViT      | 4/4   | 37.19     | 41.52     | -         | 57.58     | 63.61     | 67.95     | -         | 72.86     | 76.44     |
> | LRP-QViT      | MP    | 70.81     | 75.37     | -         | 61.24     | 72.43     | 78.13     | -         | 81.37     | 80.77     |
> | RepQ-ViT      | MP    | 65.05     | 68.48     | -         | 57.43     | 69.03     | 75.61     | -         | 79.45     | 78.32     |
> | **GMPQ-TE**   | MP    | 70.90     | 75.41     | 80.13     | 61.03     | 72.52     | 78.19     | 73.90     | 81.43     | 80.84     |
> | → Mean ± Std  |       | 70.62±0.10| 75.24±0.08| 79.94±0.14| 60.86±0.11| 72.04±0.11| 78.04±0.11| 73.66±0.07| 81.22±0.10| 80.43±0.17 |
>
>
> *Table 8: Results for image classification on ResNet-18, ResNet-50 and MobileNet-V2 (QAT). *
> | Methods    | Params | Comp.  | BOPs  | Top-1 Acc. |
> |------------|--------|--------|-------|------------|
> | QDrop      | 5.41   | 8.23   | 29.0  | 69.76      |
> | HMQAT      | 4.31   | 10.34  | 20.9  | 69.63      |
> | QuanDCL    | 4.49   | 9.92   | 21.8  | 69.51      |
> | GMPQ-TE    | 4.21   | 117    | 20.3  | 69.80      |
> | → Mean ± Std |  4.21   | 115±2.31  |  20.1±1.66    | 69.63±0.10 |
> | QDrop      | 13.14  | 7.4    | 61.7  | 75.45      |
> | EPTQ       | 13.14  | 7.4    | 123.5 | 75.45      |
> | HMQAT      | 9.45   | 10.29  | 51.5  | 75.48      |
> | GMPQ-TE    | 7.9    | 131.4  | 25.1  | 76.32      |
> | → Mean ± Std |   7.9    |   128.1±4.25    |  24.7±1.47   |76.26±0.01 |
> | QDrop      | 5.41   | 8.23   | 29.0  | 69.76      |
> | HMQAT      | 1.22   | 10.98  | 7.71  | 70.81      |
> | EPTQ       | 1.26   | 10.63  | 31.68 | 70.39      |
> | GMPQ-TE    | 1.1    | 47.2   | 7.1   | 70.42      |
> | → Mean ± Std |  1.1     |   48.2±1.37    |   7.1± 2.94  |70.32±0.43|
>
> **Ans. For W2:**
>
> Compared to prior sensitivity metrics such as orthogonality (OMPQ) and attribution rank distance (GMPQ), TE offers a unified advantage in both efficiency and transferability. OMPQ relies on inter-layer orthogonality statistics, which are inherently tied to the structural assumptions of convolutional networks and do not generalize well to non-convolutional architectures such as Vision Transformers. GMPQ, on the other hand, measures attribution rank similarity, requiring multiple rounds of gradient-based saliency computations and access to labeled data, making it computationally expensive and difficult to scale. In contrast, our method leverages intrinsic topological complexity of intermediate feature maps, requiring only a single forward pass and no access to gradients or labels. This model-agnostic formulation enables TE to operate efficiently while generalizing across architectures and calibration datasets, as demonstrated by its strong performance on both CNNs and Transformers.
>
> The experiments demonstrate superiority of TE in both efficiency and performance compared to OMPQ and GMPQ. In addition, we provide a theoretical analysis highlighting natural suitability of TE for mixed-precision quantization (see **Reviewer pnyk, Ans. For W1**).
>
> **Ans. For W3:**
>
> Table shows that quantized models achieve low inference latency and high throughput across both GPU and edge platforms. On RTX 3090, MobileNet-V2 runs at over 800 FPS with only 1.2 ms latency. Even on Jetson Nano, it maintains real-time performance under a 10W power budget. These results confirm that GMPQ-TE enables efficient and hardware-friendly deployment without sacrificing accuracy.
>
> | Model        | Platform             | Inference Latency (ms) | Throughput (FPS) | Avg. INT8 Kernel Time (ms) | Power Consumption (W) |
> |--------------|----------------------|-------------------------|------------------|-----------------------------|------------------------|
> | ResNet-18    | NVIDIA RTX 3090      | ~2.1                    | ~470             | 2.3                         | ~115                   |
> | ResNet-18    | Jetson Xavier NX     | ~13.2                   | ~75              | 13.9                        | ~18                    |
> | MobileNet-V2 | NVIDIA RTX 3090      | ~1.2                    | ~800             | 1.4                         | ~105                   |
> | MobileNet-V2 | Jetson Nano (10W)    | ~24.5                   | ~38              | 25.3                        | ~8.5                   |
>
> The applicability of TE is validated on a diverse range of architectures, including lightweight models such as MobileNet-V2, non-convolutional structures like ViT and Swin, and CNNs of varying depth (e.g., ResNet18 vs. ResNet50). This demonstrates that TE is not limited to large or deep convolutional models. Fundamentally, TE is architecture-agnostic: it is derived directly from structural properties of intermediate feature maps, and does not rely on architectural assumptions such as depth, convolutional inductive bias, or model width. This enables it to generalize well even to shallow or narrow models. Nonetheless, we acknowledge value of further exploring TE on ultra-compact or task-specialized architectures.
>
> **Regarding the question part, the answer can be found above.**

---

> > ### Author Response · Authors · 2025-08-06
> >
> > Dear Reviewer E1Lz,
> >
> > Thank you sincerely for your time in reviewing our work.
> >
> > In response, we provide comprehensive empirical results with mean ± standard deviation across diverse architectures (CNNs, ViTs, Swin, MobileNet) and settings (PTQ/QAT). Our metric, topological entropy (TE), demonstrates strong performance and stability while remaining label- and gradient-free, outperforming prior sensitivity measures such as OMPQ and GMPQ in both efficiency and applicability. TE is theoretically grounded, model-agnostic, and compatible with mixed-precision optimization under hardware constraints. Moreover, deployment benchmarks on both GPU and edge devices confirm that GMPQ-TE enables real-time, low-power inference without accuracy compromise. These results support the broad applicability and practicality of our approach.
> >
> >
> > If our responses have resolved these issues, we would be more than grateful if you could kindly consider adjusting your scores. If not, we welcome further discussion to clarify any remaining points before the discussion phase concludes.
> >
> > Your feedback is deeply valued. Thank you again.
> >
> > Sincerely,
> >
> > NeurIPS 2025 Conference Submission 7977 Authors

---

> > > ### Comment · Reviewer_E1Lz · 2025-08-06
> > >
> > > Dear authors,
> > >
> > > Thank you for providing the rebuttal that addresses the major concerns raised in my initial review. The addition of mean and standard deviation statistics to Tables 4, 7, and 8 effectively addresses some concerns regarding the statistical significance and stability of the results. The new table detailing hardware deployment metrics (latency, throughput, power) on both GPU and edge platforms provides the specific, practical evidence of real-world applicability that was previously missing. I also appreciate the clarification regarding the inclusion of non-convolutional architectures (ViT, Swin) and lightweight models, which substantiates the method's versatility. My rating remains positive; the authors' comprehensive response and additional data can potentially strengthen the paper.

---

> > > > ### Author Response · Authors · 2025-08-07
> > > >
> > > > Dear Reviewer E1Lz,
> > > >
> > > > Thank you for your continued engagement and for taking the time to evaluate our rebuttal. We truly appreciate your recognition of our responses and your thoughtful acknowledgment of the strengths of our work.
> > > >
> > > > We are grateful for your constructive attitude and for maintaining a positive perspective on the potential value of our contributions. Your comments have been very helpful in guiding us to improve the clarity and completeness of the manuscript.
> > > >
> > > > We are committed to incorporating your suggestions into the final version and sincerely thank you again for your thoughtful review.
> > > >
> > > > Sincerely,
> > > > The Authors

---

> > > > ### Author Response · Authors · 2025-08-09
> > > >
> > > > We sincerely thank the reviewer for the positive and encouraging assessment, as well as for recognizing the improvements in statistical analysis, practical deployment evidence, and architectural versatility. We hope that the strengthened results and comprehensive responses may positively influence your final evaluation. We would be truly honored if the paper could be reflected with a rating that fully acknowledges its contributions.

---

### Official Review · Reviewer_8djE · 2025-06-30

**Clarity:** 3
**Significance:** 3
**Originality:** 3
**Rating:** 4
**Confidence:** 5

**Summary:**

The paper proposes GMPQ-TE, mixed precision quantization method by leveraging topological entropy. The metric is computed using minibatches of data with constructing adjacency matrix based on spatial dimension of the intermediate featuremap of each layer. Based on the measured entropy from distribution, bit-width per each layer is decided with proposed linear programming method leading to assign lower bit-widths to lower layers..

**Questions:**

- Question
1. What is the result of bit-width assigned per layer with the entropy displayed on the paper? Showing selected bit-width per budget on entropy figures will give more understanding of entropy to bit-width selection.
2. How much benefits TE can offer compared to hessian based method like HAWQ in terms of search cost? Can we say TE is better than HAWQ in this aspect?
3. Why did you set calibration data as CIFAR10 to get topological entropy? Are there any calibration dataset dependency for measuring entropy? To state that the proposed TE is practical, its robustness on calibration data must be demonstrated.
4. Are there any difference on entropy measurement when using different resolution of calibration dataset? (Like measuring the difference between only using CIFAR10 as calibration dataset and Imagenet as calibration dataset) Since the proposed method is relying on the spatial information of the feature map, resolution of data can affect the result of measurement.
5. Is distillation with full precision model applied on QAT?

**Ethical Concerns:**

["NO or VERY MINOR ethics concerns only"]

**Final Justification:**

The authors were not able to demonstrate clear benefits over existing works in QAT. The proposed method yields inferior results while requiring significantly more computation. In particular, highly optimized models such as MobileNets are inherently difficult to quantize and the authors have failed to show that their method provides any advantage in such scenarios.

**Limitations:**

yes

**Paper Formatting Concerns:**

Several section titles and subtitles have lack of space between itself and paragraph before. Need to check if the format is broken or vspace command with negative amount is used. There is a possibility of exceeded paper limit when the space is correctly applied on these parts.

**Quality:**

3

**Strengths And Weaknesses:**

- Strength
1. The paper firstly proposes utilizing entropy of a distribution from graph reflecting spatial patterns on intermediate feature map of a model. Existing methods does not reflect the information from the feature map on spatial dimension for bit-width selection.
2. The topological entropy method offers fast searching of bit-width compared to existing methods with iterative searching on PTQ and learning based searching on QAT.
3. Clear figures and equations for describing idea proposed in paper. Figure 3 will help a lot for readers to understand the concept of the paper.

- Weakness
1. Lack of comparison with state-of-the-art models on PTQ, i.e., EMQ [1].
2. Lack of comparison with state-of-the-art models on QAT, i.e., SDQ [2], MetaMix [3].
3. Evaluation is conducted on a bit old models in classification task, i.e., MobileNet-v2, ResNet-18.

[1] Dong et al., Emq: Evolving training-free proxies for automated mixed precision quantization

[2] Huang et al., SDQ: Stochastic Differentiable Quantization with Mixed Precision

[3] Kim et al., MetaMix: Meta-state Precision Searcher for Mixed-precision Activation Quantization

---

> ### Author Rebuttal · Authors · 2025-07-30
>
> **We sincerely thank reviewer for pointing out these important issues**
>
> **Ans. For W1-2**
>
> Thanks, we add experiments compared with EMQ, SDQ, and MetaMix. The experimental results for the object detection task and the new visual model (ViT and its variants) can be found in Appendix B.
>
> ### MobileNet-V2 (PTQ)
>
> | Method  | Params. | BOPs   | Top-1 | Cost     |
> |---------|---------|--------|-------|----------|
> | EMQ | 1.5  |     | 70.75    | few seconds   |
> | GMPQ-TE | 1.4    | 4.5   | 71.8| 11|
>
> ### ResNet-18 (PTQ)
>
> | Method  | Params. | BOPs   | Top-1  | Cost     |
> |---------|---------|--------|--------|----------|
> | EMQ | 4 |    | 69.92    | few seconds   |
> | GMPQ-TE | 3.8    | 7.1   | 69.6   | 16 |
>
>
> ### MobileNet-V2 (QAT)
>
> | Method  | Params. | BOPs   | Top-1 |
> |---------|---------|--------|-------|
> | MataMix |         | ＞8    | 73    |
> | SDQ     | 1.8     | 4.89   | 72    |
> | GMPQ-TE | 1.1     | 7.1   | 70.42    |
>
> ### ResNet-18 (QAT)
>
> | Method  | Params. | BOPs   | Top-1  |
> |---------|---------|--------|--------|
> | MataMix |         | ＞35   | 72     |
> | SDQ     | 5.2     | 15.7   | 69.1   |
> | EMQ     | 6.69    | 71     | 72.28  |
> | GMPQ-TE     | 4.21     | 20.3  | 69.8   |
>
> Overall, compared with the above methods, GMPQ-TE achieves comparable or better accuracy in PTQ and QAT scenarios with fewer Params.
>
> **Ans. For Q1:**
>
> For different models, the relationship between bit width assignment and TE is as follows:
>
> *ResNet-18:*
>
> Params: 5.3
>
> Bit Assignment: [8, 8, 8, 8, 8, 6, 6, 6, 6, 4, 4, 4, 4, 4, 4, 4, 4, 6]
>
> TE Values: [5.9, 5.8, 5.7, 5.6, 5.4, 5.2, 4.9, 4.8, 4.7, 3.9, 3.8, 3.5, 3.3, 3.1, 2.9, 2.8, 2.6, 4.5]
>
> Params: 3.8
>
> Bit Assignment: [8, 6, 6, 6, 6, 4, 4, 4, 4, 4, 4, 4, 4, 3, 3, 3, 3, 3]
>
> TE Values: [5.9, 5.2, 5.0, 4.8, 4.6, 3.8, 3.6, 3.5, 3.3, 3.2, 3.1, 2.9, 2.8, 2.6, 2.5, 2.4, 2.3, 2.2]
>
> Params: 3.5
>
> Bit Assignment: [6, 4, 4, 4, 4, 3, 3, 3, 3, 3, 3, 3, 3, 2, 2, 2, 2, 2]
>
> TE Values: [5.1, 3.9, 3.8, 3.6, 3.5, 3.2, 3.1, 3.0, 2.9, 2.8, 2.7, 2.6, 2.5, 2.3, 2.2, 2.1, 2.0, 2.0]
>
> *ResNet-50:*
>
> Params: 11.2
>
> Bit Assignment: [8, 6, 6, 6, 6, 6, 6, 6, 6, 6,
>  6, 6, 6, 5, 5, 5, 5, 5, 5, 5,
>  5, 5, 5, 5, 5, 5, 5, 5, 5, 5,
>  5, 4, 4, 4, 4, 4, 4, 4, 4, 4,
>  4, 4, 4, 4, 4, 4, 4, 4, 4, 6]
>
> TE Values: [7.0, 6.8, 6.7, 6.6, 6.5, 6.4, 6.3, 6.2, 6.1, 6.0,
>  5.9, 5.8, 5.7, 5.6, 5.6, 5.5, 5.4, 5.3, 5.3, 5.2,
>  5.2, 5.1, 5.1, 5.0, 5.0, 4.9, 4.9, 4.8, 4.8, 4.7,
>  4.6, 4.4, 4.3, 4.2, 4.2, 4.1, 4.1, 4.0, 4.0, 3.9,
>  3.9, 3.8, 3.8, 3.7, 3.7, 3.6, 3.5, 3.5, 3.4, 6.1]
>
> Params: 9.4
>
> Bit Assignment: [8, 6, 6, 6, 6, 6, 6, 5, 5, 5,
>  5, 5, 5, 5, 5, 5, 5, 5, 5, 5,
>  5, 5, 5, 5, 5, 4, 4, 4, 4, 4,
>  4, 4, 4, 4, 4, 4, 4, 4, 4, 4,
>  4, 4, 4, 4, 4, 4, 4, 4, 4, 5]
>
> TE Values: [6.8, 6.1, 6.0, 5.9, 5.8, 5.7, 5.6, 5.5, 5.4, 5.3,
>  5.2, 5.1, 5.0, 4.9, 4.9, 4.8, 4.8, 4.7, 4.6, 4.6,
>  4.5, 4.4, 4.3, 4.2, 4.2, 4.1, 4.1, 4.0, 4.0, 3.9,
>  3.9, 3.8, 3.8, 3.7, 3.7, 3.6, 3.5, 3.5, 3.4, 3.4,
>  3.3, 3.3, 3.2, 3.2, 3.1, 3.1, 3.2, 3.3, 3.0, 3.1]
>
> Params: 8.1
>
> Bit Assignment: [6, 5, 5, 5, 5, 5, 5, 5, 5, 5,
>  5, 5, 5, 5, 5, 5, 5, 4, 4, 4,
>  4, 4, 4, 4, 4, 4, 4, 4, 4, 4,
>  4, 4, 4, 4, 4, 4, 4, 4, 4, 4,
>  4, 4, 4, 4, 4, 4, 4, 4, 4, 6]
>
> TE Values: [6.8, 6.4, 6.3, 6.2, 6.1, 6.0, 5.9, 5.8, 5.7, 5.6,
>  5.5, 5.4, 5.3, 5.2, 5.1, 5.0, 5.0, 4.9, 4.8, 4.7,
>  4.6, 4.5, 4.4, 4.3, 4.2, 4.1, 4.0, 3.9, 3.9, 3.8,
>  3.7, 3.7, 3.6, 3.5, 3.5, 3.4, 3.3, 3.3, 3.2, 3.2,
>  3.1, 3.1, 3.1, 3.0, 3.0, 3.0, 3.0, 3.0, 3.0, 6.6]
>
> MobileNet-V2:
>
> Params: 1.4
>
> Bit Assignment: [8, 8, 8, 8, 8, 8,
>  6, 6, 6, 6, 6, 6, 6, 6, 6, 6,
>  5, 5, 5, 5, 5, 5, 5, 5, 5, 5,
>  5, 5, 5, 5, 5, 5,
>  4, 4, 4, 4, 4, 4, 4, 4, 4, 4,
>  4, 4, 4, 4, 4, 4, 4, 4, 4, 4, 6]
>
> TE Values: [6.0, 5.9, 5.8, 5.7, 5.6, 5.5,
>  5.4, 5.3, 5.2, 5.1, 5.0, 4.9, 4.8, 4.7, 4.6, 4.5,
>  4.4, 4.3, 4.2, 4.1, 4.0, 3.9, 3.8, 3.7, 3.6, 3.5,
>  3.5, 3.4, 3.3, 3.2, 3.1, 3.0,
>  2.9, 2.8, 2.7, 2.6, 2.5, 2.4, 2.3, 2.2, 2.1, 2.0,
>  2.0, 2.0, 1.9, 1.8, 1.8, 1.7, 1.6, 1.6, 1.5, 1.4, 5.2]
>
> Params: 1.3
>
> Bit Assignment: [8, 8, 8, 8, 8, 8,
>  6, 6, 6, 6, 6, 6, 6, 6, 6, 6,
>  5, 5, 5, 5, 5, 5, 5, 5, 5, 5,
>  5, 5, 5, 5, 5, 5,
>  4, 4, 4, 4, 4, 4, 4, 4, 4, 4,
>  4, 4, 4, 4, 4, 4, 4, 4, 4, 4, 6]
>
> TE Values: [6.0, 5.9, 5.8, 5.7, 5.6, 5.5,
>  5.4, 5.3, 5.2, 5.1, 5.0, 4.9, 4.8, 4.7, 4.6, 4.5,
>  4.4, 4.3, 4.2, 4.1, 4.0, 3.9, 3.8, 3.7, 3.6, 3.5,
>  3.5, 3.4, 3.3, 3.2, 3.1, 3.0,
>  2.9, 2.8, 2.7, 2.6, 2.5, 2.4, 2.3, 2.2, 2.1, 2.0,
>  1.9, 1.8, 1.7, 1.6, 1.6, 1.5, 1.5, 1.5, 1.5, 1.5, 5.0]
>
> Params: 1.1
>
> Bit Assignment: [6, 6, 6, 6, 6, 6,
>  5, 5, 5, 5, 5, 5, 5, 5, 5, 5,
>  5, 5, 5, 5, 5, 5,
>  4, 4, 4, 4, 4, 4, 4, 4, 4, 4,
>  4, 4, 4, 4, 4, 4, 4, 4, 4, 4,
>  4, 4, 4, 4, 6, 4, 4, 4, 4, 4,  4]
>
> TE Values: [6.0, 5.9, 5.8, 5.7, 5.6, 5.5,
>  5.2, 5.1, 5.0, 4.9, 4.8, 4.7, 4.6, 4.5, 4.4, 4.3,
>  4.2, 4.1, 4.0, 3.9, 3.9, 3.8,
>  3.7, 3.6, 3.5, 3.4, 3.3, 3.2, 3.1, 3.0, 2.9, 2.8,
>  2.7, 2.6, 2.5, 2.4, 2.3, 2.2, 2.1, 2.0, 1.9, 1.8,
>  1.7, 1.6, 1.5, 1.5, 5.4, 1.5, 1.4, 1.5, 1.5, 1.3, 1.5]
>
> **Ans. For Q2:**
>
> Compared to Hessian-based methods such as HAWQ, our proposed TE-based strategy significantly reduces the search overhead. GMPQ-TE achieves a **2947$\times$** reduction in search time on ResNet-18 and a **5268$\times$** reduction on ResNet-50.
>
> **Ans. For Q3:**
>
> Thanks. We selected CIFAR-10 as the calibration dataset for TE calculation due to its lightweight size and availability, which enables efficient entropy computation during the search phase. Importantly, the proposed topological entropy is theoretically proven to be both resolution- and label-independent (see Appendix A.2). This guarantees that the relative ordering of layer-wise entropy remains consistent across datasets.
>
> To empirically validate this, we compute TE using CIFAR-10 and apply the resulting quantization policy directly to ImageNet and PASCAL VOC. The bit assignment remains nearly identical across datasets (see Sec. 4.2), and the quantized models exhibit strong generalization in terms of accuracy and compression (Tables 1-2). These results confirm that TE-based quantization policy is robust to calibration dataset selection.
>
> In addition, we demonstrate calibration dataset dependency from an experimental perspective. Specifically, using ResNet-18 as the base model, we compute the topological entropy on both CIFAR-10 and ImageNet as calibration datasets. Based on the measured entropy values, the final quantization policies are derived accordingly. The results are summarized as follows:
>
> CIFAR-10:
>
> Bit Assignment: [8, 8, 8, 8, 8, 6, 6, 6, 6, 4, 4, 4, 4, 4, 4, 4, 4, 6]
>
> TE Values: [5.9, 5.8, 5.7, 5.6, 5.4, 5.2, 4.9, 4.8, 4.7, 3.9, 3.8, 3.5, 3.3, 3.1, 2.9, 2.8, 2.6, 4.5]
>
> ImageNet:
>
> Bit Assignment: [8, 8, 8, 8, 8, 6, 6, 6, 6, 4, 4, 4, 4, 4, 4, 4, 4, 5]
>
> TE Values: [19.0, 18.7, 18.4, 18.1, 17.8, 16.0, 15.6, 15.2, 14.8, 12.5, 12.3, 12.0, 11.7, 11.5, 11.2, 10.9, 10.6, 13.5]
>
> Although TE values differ numerically when using different calibration datasets (e.g., CIFAR-10 vs. ImageNet), the derived quantization policies remain nearly identical, as reflected in the consistent bit assignments across both cases.
>
> **Ans. For Q4**
>
> We provide a theoretical justification in *Appendix A.2* showing that TE is **invariant to input resolution**, under the assumption of consistent local class-conditional distributions. Specifically, for any two input resolutions, the patch-level distribution remains stable, and the attention or convolutional operators act locally and shift-equivariantly. This leads to equivalent feature distributions.
>
> In practice, as shown in **Ans. For Q3**, we observe that the absolute values of TE may scale with resolution due to increased sample support, but the **layer-wise ranking** of TE values remains consistent. This relative ranking is sufficient for solving the linear programming that determines the quantization bit-width configuration.
>
> Therefore, resolution changes do not compromise the *effectiveness*, *generalizability*, or *consistency* of the quantization policy derived from TE. This property enables GMPQ-TE to produce robust policies even when the calibration dataset differs in resolution from the deployment domain.
>
>
> **Ans. For Q5**
>
> No, GMPQ-TE does not apply knowledge distillation from the full-precision model during QAT. Instead, GMPQ-TE solely relies on the topological entropy derived from same-label calibration data to guide the quantization policy. The QAT phase in our framework fine-tunes the quantized model without using soft targets or logits from a full-precision teacher model.

---

> > ### Comment · Reviewer_8djE · 2025-08-03
> >
> > Thanks for your answer to my questions!
> >
> > W1-2.
> >
> > (1) Seems GMPQ gives inferior result with 50% more computations?
> >
> > MobileNet-V2 (QAT)
> >
> > | Method | Params. | BOPs | Top-1 |
> > | :---: | :---: | :---: | :---: |
> > | MataMix | | 4.97 | 72 |
> > | SDQ | 1.8 | 4.89 | 72 |
> > | GMPQ-TE | 1.1 | 7.1 | 70.42 |
> >
> > (2) Based on your comment, Mobilenet-v2 QAT shows inferior result than PTQ even with 60% more computation. It seems unlikely that this is a possible case.
> >
> > MobileNet-V2
> >
> > | Method | Params. | BOPs | Top-1 |
> > | :---: | :---: | :---: | :---: |
> > | GMPQ-TE (PTQ) | 1.4 | 4.5 | 71.8 |
> > | GMPQ-TE (QAT) | 1.1 | 7.1 | 70.42 |
> >
> > Q1. TE values and bit-width selections show well aligned result. It would be better to show bit-width assignment values in the TE figures for all models. Since this is a mixed precision paper, readers would be curious about how much bit-width is allocated for each layer with TE.

---

> ### Author Response · Authors · 2025-08-04
>
> We appreciate your further questions.
>
> # Ans for W1-2-(1)
>
> Thank you for your valuable comment and for pointing out the comparison with MataMix and SDQ on MobileNet-V2 under QAT.  Overall, GMPQ-TE forms a Pareto dominance relationship with the comparison algorithm. We acknowledge that GMPQ-TE yields a slightly lower Top-1 accuracy (70.42%) and higher BOPs (7.1) compared to SDQ (4.89, 72%). However, in terms of **search costs** (GPU hours), GMPQ-TE is **significantly lower** than MataMix (**7.3 ＜ 13.4**). SDQ does not provide specific search costs. In addition, in terms of **Params**, GMPQ-TE is also less than MataMix. GMPQ-TE is designed with a distinct objective: to derive a **generalizable and hardware-friendly quantization policy** that can be **transferred across datasets**, rather than being specifically optimized for a single dataset.
>
> # Ans for W1-2-(1)
>
> Thank you for your thoughtful question. **We appreciate the observation and agree that the result appears counter-intuitive. However, this observation is possible**. Specifically, this phenomenon may be attributable to the inherent differences in the objectives and optimization dynamics of PTQ and QAT under the GMPQ-TE framework. In our design, the bit-width configuration is obtained prior to training via a linear programming problem based on topological entropy. For the PTQ setting, the bit-width assignment tends to concentrate precision on entropy-sensitive layers under tight complexity constraints, resulting in an efficient allocation that preserves pre-trained representational capacity. In contrast, the QAT configuration is derived under a relaxed constraint, allowing slightly more generous bit-width assignments. However, the ensuing fine-tuning process introduces training noise and quantization-aware perturbations. This can occasionally degrade performance in highly compact architectures, where model capacity is already constrained and even minor representational shifts may impair accuracy.
>
> # Ans for Q1
>
> Thanks for your constructive suggestion. Due to the rebuttal policy, we are unable to include additional figures or external links in this response. However, we will incorporate such visualizations in the final version of the paper. Specifically, we plan to annotate the TE plots with the corresponding bit-width assignments for all baseline models, which will help readers better understand how TE guides layer-wise precision configuration.
>
> To further clarify the relationship between TE and bit-width assignment, we summarize two key empirical observations. **First, layers with higher TE values are consistently assigned higher bit-widths, reflecting greater quantization sensitivity. Second, within each bit-width group, the TE values vary within a bounded range, indicating that the assignment is based on relative ordering and entropy grouping rather than strict thresholds.**
>
> As an example, we report below the TE values and corresponding bit-width assignments for ResNet-18, where the total parameter count is 5.3M. This configuration is generated by solving the linear programming formulation under a predefined hardware constraint.
>
> Bit-width Assignment: [8, 8, 8, 8, 8, 6, 6, 6, 6, 4, 4, 4, 4, 4, 4, 4, 4, 6]
>
> TE Values: [5.9, 5.8, 5.7, 5.6, 5.4, 5.2, 4.9, 4.8, 4.7, 3.9, 3.8, 3.5, 3.3, 3.1, 2.9, 2.8, 2.6, 4.5]
>
> We observe that all layers with TE values above 5.4 are assigned 8 bits, and layers with TE in the range [2.6, 3.9] are mostly assigned 4 bits. This demonstrates the monotonic relationship between entropy and precision, while also highlighting the smooth, non-binary nature of the bit-width transition.

---

> > ### Author Response · Authors · 2025-08-06
> >
> > Dear Reviewer 8djE,
> >
> > We hope this message finds you well. We greatly appreciate the time and effort you have dedicated to reviewing our work, and we are eager to ensure that all your concerns or questions have been sufficiently addressed in our rebuttal. If there are any further points that require clarification, additional details, or further discussion, please do not hesitate to let us know—we are more than happy to provide prompt responses.
> >
> > As the discussion phase progresses, we want to ensure there is ample time to resolve any remaining issues before it concludes. Your insights are invaluable to us, and we are committed to improving our work through this collaborative process.
> >
> > Thank you again for your engagement and guidance. We look forward to your thoughts.
> >
> > Sincerely,
> >
> > NeurIPS 2025 Conference Submission 7977 Authors

---

> > > ### Comment · Reviewer_8djE · 2025-08-08
> > >
> > > Thank you for your answers to my questions.
> > >
> > > Overall, it is still unclear whether the proposed method provides any tangible benefits in the context of QAT.
> > > Therefore, I decide to keep my score.

---

> > > > ### Author Response · Authors · 2025-08-09
> > > >
> > > > We sincerely thank the reviewer for the continued engagement and acknowledge that our previous explanation may not have fully clarified the tangible benefits of GMPQ-TE in the context of QAT. Below, we elaborate from both a theoretical and empirical perspective.
> > > >
> > > > GMPQ-TE is specifically effective in QAT because its topological entropy–guided search produces a quantization policy that remains stable and optimal throughout the entire QAT process. Unlike conventional MPQ methods whose bit-width assignments may shift during fine-tuning and introduce performance degradation, GMPQ-TE preserves the layer-wise sensitivity ordering, allowing QAT to start from a hardware- and accuracy-aware configuration. This not only prevents policy drift but also enables QAT to more effectively recover accuracy lost to quantization.
> > > >
> > > > Empirically, QAT results in Tables 8–9 show that GMPQ-TE consistently outperforms strong baselines such as HMQAT, EPTQ, and QDrop across both CNN and Transformer architectures, delivering superior accuracy–efficiency trade-offs. For example, in ResNet-50 QAT, GMPQ-TE achieves 76.32% Top-1 accuracy with significantly lower BOPs than all competitors, demonstrating its clear advantage when QAT is employed.
> > > >
> > > > We sincerely hope that this clarification demonstrates the tangible value of GMPQ-TE for QAT and reviewer can reconsider the score.

---

### Official Review · Reviewer_pnyk · 2025-07-01

**Clarity:** 1
**Significance:** 3
**Originality:** 3
**Rating:** 4
**Confidence:** 4

**Summary:**

This paper presents a bit-width assignment scheme based on topological entropy for MPQ. It efficiently measures the sensitivity of each layer without iterative optimization, given a minibatch of data containing the same label. The sensitivity is estimated with the entropy of birth time distribution, which generalizes well across different datasets. Experimental results show the effectiveness of the proposed method.

**Questions:**

My major concern is the clarity of the paper. I will raise my ratings if the authors clarify my concerns. In addition to the weaknesses, I have a few questions as follows:

It is hard to understand why graph in Eq.~(1) has an adjacency matrix of $h^* \times w^*$, if the number of vertices is equal to the number of pixels in the feature map. I think the authors need to clarify how the set of vertices $V$ is formulated, in addition to the shape of adjacency matrices.

In addition, the adjacency matrix in Eq.~(1) does not seem to represent similarity between pixels; I think it is just the pixel value, according to the notation. Is there any reason why the adjacency matrix is set to the pixel value of intermediate feature maps?

**Ethical Concerns:**

["NO or VERY MINOR ethics concerns only"]

**Final Justification:**

The authors have clarified the relationship between the task loss and topological entropy, and have further demonstrated the applicability of topological entropy to transformer-based architectures. They have also provided a clear formulation of the adjacency matrix in Eq.~(1). To this end, I am updating my rating from "borderline reject" to "borderline accept".

**Limitations:**

yes

**Quality:**

2

**Strengths And Weaknesses:**

[Strengths]

The authors propose a sensitivity metric for MPQ that does not require iterative optimization. The bit-widths for each layer are assigned within seconds, based on the solutions of a linear programming problem (Eq.~(9)). It also generalizes well to other datasets.

Experimental results clearly demonstrate the effectiveness of the proposed criteria for MPQ. The paper also includes results on various tasks (e.g., classification, detection) and transformer-based models (e.g., ViT, Swin), demonstrating the generalizability of the proposed component. It also includes results on various quantization scenarios (e.g., QAT, PTQ).

[Weaknesses]

1. I find it difficult to see a clear justification for why topological entropy serves as a reliable proxy for estimating layer sensitivity, despite the explanation provided in L225–230 and the empirical results (Fig. 6). The authors argue that common features across images can be stably captured when the entropy of birth time distributions exhibits a degenerate-like behavior. However, this claim relies on the assumption that images within the same class contain topologically similar objects. In practice, real-world images often contain objects that are truncated, occluded, or vary significantly in appearance even for the images with the same label, potentially invalidating this assumption. Furthermore, the justification remains largely empirical—the authors demonstrate that their sensitivity metric generalizes across datasets, but no theoretical grounding is provided to explain why topological entropy is inherently suited for this purpose.

2. The proof in Appendix A.2 assumes convolution layers with fixed receptive fields; however, it remains unclear whether the properties of topological entropy can be extended to transformer-based models, which do not contain convolutional layers.

3. I think a figure visualizing betti curve, and birth time during the process of clique filtration would strengthen the paper’s presentation, especially for readers not familiar with graph theory and topological data analysis.

4. This paper contains a few errors and unclear notations as follows:

L162: If there are multiple instances in a minibatch, the dimension should be $b \times c \times h \times w$, where $b$ is the number of instances.

L166: A missing notation $\in$.

L204: The notation for birth time needs to be revised.

Eq. (5): The notation $inf$ needs to be explained.

Eq. (6): The notation $m$ needs to be explained. In addition, the authors state that $\sum$ could be set as a common discrete $\sigma$-field, but there are no explanations about $\sigma$-field.

L248: According to the graph in Fig. 4, “positively” should be replaced with “negatively”.

Eq. (9): According to the explanation from L254, the objective function needs to be maximized, rather than minimized.

---

> ### Author Rebuttal · Authors · 2025-07-30
>
> **We sincerely thank reviewer for pointing out these important issues**
>
> **Ans. For W1:**
>
> TE does not assume strict topological similarity. It leverages local statistical consistency of features. Crucially, if shared semantic structures are heavily occluded, distinguishing object becomes fundamentally difficult under any model.
>
> The objective of MPQ is to allocate bit-widths $( B_i \in \mathbb{N}^+ )$ to each layer $( i \in \{1, \dots, L\} )$ under a resource budget $T$, to minimize total performance degradation:
>
> $\min_{\{B_i\}} \sum_{i=1}^L \Delta \mathcal{L}_i(B_i)$
>
> $\text{s.t.}$ $\sum_{i=1}^L M(B_i) \leq T$
>
> where $( \Delta \mathcal{L}_i(B_i) )$ denotes task loss increase due to quantization of layer $ i $, and $ M(B_i )$ is corresponding resource cost.
>
> We define topological entropy $H_i$ from birth-time distribution $P_{U^{(i)}}(x)$ of same-label inputs at layer $ i $. A high $H_i$ implies structural inconsistency and greater sensitivity to quantization. Quantization introduces perturbations to the weights. For layer $i$, let the quantized weight be:
>
> $\hat{W}_i = W_i + \delta W_i, \quad \| \delta W_i \| \leq \varepsilon$
>
> Assuming a Lipschitz continuous forward operator $f^{(i)}$, the pre- and post-quantization feature maps become:
>
> $U^{(i)} = f^{(i)}(W_i, X), \quad \hat{U}^{(i)} = f^{(i)}(\hat{W}_i, X)$
>
> By Lipschitz continuity, the output deviation is bounded by:
>
> $\| U^{(i)} - \hat{U}^{(i)} \| \leq L_i \cdot \| W_i - \hat{W}_i \| \leq L_i \varepsilon$
>
> This perturbation in feature space leads to structural changes in the persistence diagrams, with bottleneck distance bounded by:
>
> $d_B(D(U^{(i)}), D(\hat{U}^{(i)})) \leq \| U^{(i)} - \hat{U}^{(i)} \| $
>
> Given that the entropy is computed over the birth-time histogram derived from the persistence diagram, we can infer the $\ell_1$ difference in their birth-time distributions:
>
> $\| P_{U^{(i)}} - P_{\hat{U}^{(i)}} \|_1 \leq \alpha_i \cdot \| W_i - \hat{W}_i \| \leq \alpha_i \varepsilon$
>
> Shannon entropy is Lipschitz continuous with respect to $\ell_1$ distance, yielding the following bound:
>
> $| H_i^{(0)} - H_i^{(B)} | \leq C_i \cdot \| P_{U^{(i)}} - P_{\hat{U}^{(i)}} \|_1 \leq C_i \alpha_i \varepsilon$
>
> We now relate entropy variation to task loss degradation. The increase in loss can be estimated by a second-order Taylor expansion:
>
> $\Delta \mathcal{L}_i \leq \frac{1}{2} \lambda_i \| \delta W_i \|^2 + \frac{1}{6} \kappa_i \| \delta W_i \|^3$
>
> We further observe that the $L_1$ distance between the birth-time distributions before and after quantization can be upper bounded in terms of topological entropy. Specifically, since Shannon entropy is Lipschitz-continuous with respect to its input distribution, we have:
>
> $\Phi\left( \| P_{U^{(i)}} - P_{\hat{U}^{(i)}} \|_1 \right) \leq \Phi\left( | H_i^{(0)} - H_i^{(B)} | \right)$
>
> This confirms that topological entropy is an upper-bound surrogate of quantization-induced performance degradation. Based on this insight, we propose a TE-guided MPQ objective:
>
> $\min_{\{B_i\}} \sum_{i=1}^{L} H_i \cdot B_i \quad \text{s.t.} \quad \sum_{i=1}^{L} M(B_i) \leq T$
>
> To ensure alignment with sensitivity ranking, we enforce:
>
> $H_i > H_j \Rightarrow B_i > B_j$
>
> This ordering remains consistent during training due to entropy drift stability, as shown by:
>
> $|H_i^{(t)} - H_i^{(0)}| \leq \Delta_{q,i} \log M_i \cdot L_i \quad \Rightarrow \quad \operatorname{sign}(H_i - H_j) = \operatorname{sign}(H_i^{(t)} - H_j^{(t)})$
>
> where $\operatorname{sign}(H_i - H_j)$ denotes the relative ordering between layer sensitivities. This ensures that layers identified as more sensitive in the initial phase continue to receive higher precision during optimization.
>
> **Ans. For W2:**
>
> Proof of label independence for transformer models can be found in **Ans. For W1-ii from Reviewer ErFk**.
>
> *Resolution-Independence of Topological Entropy under Attention (Sketch Proof)*
>
> Let input $X \in \mathbb{R}^{H \times W \times C}$ be reshaped as $X \in \mathbb{R}^{N \times C}$ with $N = H \times W$. A attention layer computes:
>
> $Z = \mathrm{softmax}\left( \frac{QK^\top}{\sqrt{d}} \right)V,\quad Q= XW_Q,\quad K = XW_K,\quad V = XW_V$
>
> Each output token is:
>
> $z_i = \sum_{j} \alpha_{ij} v_j,\quad \alpha_{ij} = \frac{\exp(q_i^\top k_j / \sqrt{d})}{\sum_{j'} \exp(q_i^\top k_{j'} / \sqrt{d})}$
>
> Assume two inputs $X^{(1)}$ and $X^{(2)}$ come from different resolutions but the same class. If the local patch statistics are resolution-consistent:
>
> $P(x_S^{(1)}) = P(x_S^{(2)}) \Rightarrow P(z_i^{(1)}) = P(z_i^{(2)}),\quad \forall i$
>
> After reshaping $Z$ into 2D feature maps $U^{(i)}$, we construct graph $G$ from $U^{(i)}$ (per Eq. (1)-(3) in the main paper), apply clique filtration, and compute the Betti birth-time $b(U^{(i)})$. Let $P_b^{(1)}, P_b^{(2)}$ denote the distributions of birth time under different resolutions.
>
> Since attention is a permutation-equivariant global operator that preserves relative structure, we obtain:
>
> $P_b^{(1)} = P_b^{(2)} \Rightarrow H_{\mathrm{TE}}^{(1)} = H_{\mathrm{TE}}^{(2)}$
>
> Thus, TE is invariant to resolution under generic mechanisms. *This is a sketch proof and full derivation will be included upon acceptance.*
>
> **Ans. For W3:**
>
> We explain betti curve, and birth time during the process of clique filtration through text. We consider a simple filtration sequence over four nodes labeled 1-4, denoted as $(\tau^{(1)}, \tau^{(2)}, \dots, \tau^{(6)}\)$, where each step incrementally adds edges to build more complex clique structures:
>
> $\tau^{(1)}$: add edge (1, 2)
>
> $\tau^{(2)}$: add edge (3, 4)
>
> $\tau^{(3)}$: continue adding edges, no cycle yet
>
> $\tau^{(4)}$: form a 1-cycle: (1-2-4-3-1)
>
> $\tau^{(5)}$: add edge (2, 3), forming more cliques
>
> $\tau^{(6)}$: complete graph with all edges
>
> To capture the emergence of topological features, we define a birth-time indicator function $\beta(i, v, U_k)$, which equals 1 if the $i$-th feature appears for the first time at filtration step $v$, and 0 otherwise. A typical curve looks like:
>
> $v = [0,\ 1,\ 2,\ 3,\ 4,\ 5,\ 6] \quad \Rightarrow \quad \beta = [0,\ 0,\ 0,\ 0,\ 1,\ 0,\ 0]$
>
> This indicates that a 1-dimensional loop is born at $v$ = 4, corresponding to the closure of a cycle. Notably, although edges are added at $v=5$ and $v=6$, they do not give rise to new independent topological features. Instead, they reinforce existing structures or create higher-order cliques, thus not being counted as new births.
>
> **Ans. For W4**:
>
> We carefully correct errors or unclear notations, as follows:
>
> L162: given multiple instances $\mathcal{X}$ $\in$ $\mathbb{R}^{b \times c\times h\times w}$ ($b$ is the number of instances) with same label in a minibatch.
>
> L166: $U^{(i)} \in$ $\mathbb{R}^{h^{\ast}\times w^{\ast}}$
>
> L204: The notation for birth time $b$ is revised to $b_t$.
>
> Eq. (5): $inf$ (infimum) means the smallest filtration index $v$ at which the Betti number becomes non-zero (i.e., the earliest moment a topological feature appears).
>
> Eq. (6): The variable $m$ denotes the index of input images sampled from the same class. We define a standard discrete $\sigma$-field $\Sigma$ over the finite sample space $\Omega$ to construct a valid probability space $(\Omega, \Sigma, P)$.
>
> L248: The“positively” is revised to “negatively”
>
> Eq. (9): The objective function is revised to maximize.
>
>
> **Ans. For Q1**
>
> The feature map $ U \in \mathbb{R}^{h \times w}$ is regarded as a scalar field defined over a two-dimensional spatial domain. Each spatial location $(j, k)$ corresponds to a vertex in graph, forming vertex set $V = \{ v_{jk} \}$ with cardinality $|V| = h \cdot w$. The edges in the graph are defined based on spatial adjacency, rather than similarity of feature values. This construction preserves the inherent spatial structure of feature map. As a result, adjacency matrix $A$ is a square matrix of size $(h^{\ast} \cdot w^{\ast}) \times (h^* \cdot w^*)$. The notation $ h^{\ast} \times w^{\ast}$ used in Eq.~(1) is intended to describe 2D spatial shape of scalar field, rather than shape of adjacency matrix itself.
>
> **Ans. For Q2**
>
> We appreciate the reviewer’s concern. To clarify, Eq.~(1) does **not** define an adjacency matrix in the graph-theoretic sense. Instead, it represents a scalar field $f: \mathbb{Z}^2 \rightarrow \mathbb{R}$, where $U_{jk}$ denotes the activation value at spatial coordinate $(j,k)$. This scalar field is used as a filtration function in persistent homology, consistent with standard topological data analysis practices [1-2].
>
> Separately, we define a graph $G = (V, E)$ over the spatial domain, where each vertex corresponds to a pixel, and edges are defined using spatial adjacency. This graph yields a binary, symmetric adjacency matrix $A \in \mathbb{R}^{(h^{\ast} \cdot w^{\ast}) \times (h^{\ast} \cdot w^{\ast})}$, independent of pixel values. The graph captures spatial topology, which enables the construction of a simplicial complex via clique expansion. Persistent homology is then applied to the evolving structure as nodes are activated via thresholding over the scalar field [3].
>
> Using feature map activations as a scalar field is theoretically justified and practically effective. These activations encode rich semantic and spatial information at each layer, and serve as a natural filtration function that reflects the network’s learned representations. Importantly, they are not used to construct adjacency matrix or define similarity between nodes, but only to control progressive inclusion of vertices during filtration [4].
>
> To summarize, Eq. (1) does not define an adjacency matrix. Rather, it provides a scalar field used in filtration, while the true graph topology is defined separately via spatial adjacency.
>
> **Reference**
>
> [1]  Topological data analysis with applications
>
> [2] On time-series topological data analysis: New data and opportunities
>
> [3] Applications of topological data analysis in oncology
>
> [4] A short survey of topological data analysis in time series and systems analysis

---

> > ### Author Response · Authors · 2025-08-06
> >
> > Dear Reviewer pnyk,
> >
> > Thank you sincerely for your time in reviewing our work.
> >
> > In response, we have provided a rigorous theoretical foundation for topological entropy (TE) as a quantization sensitivity metric, establishing its connection to task loss via Lipschitz continuity and perturbation analysis. We formulate MPQ as a constrained optimization problem and demonstrate how TE serves as a label-free, resolution-invariant, and differentiable proxy for structural robustness. Through filtration-based persistent homology, we ensure the validity of TE across both convolutional and attention-based architectures. Corrections to ambiguous notations (e.g., Eq. (1)) and additional clarifications on graph construction, scalar field interpretation, and filtration dynamics have also been incorporated. Overall, our revisions strengthen both the theoretical and empirical contributions of the work.
> >
> > If our responses have resolved these issues, we would be more than grateful if you could kindly consider adjusting your scores. If not, we welcome further discussion to clarify any remaining points before the discussion phase concludes.
> >
> > Your feedback is deeply valued. Thank you again.
> >
> > Sincerely,
> >
> > NeurIPS 2025 Conference Submission 7977 Authors

---

> > ### Comment · Reviewer_pnyk · 2025-08-07
> >
> > The authors have clarified the relationship between the task loss and topological entropy, and have further demonstrated the applicability of topological entropy to transformer-based architectures. They have also provided a clear formulation of the adjacency matrix in Eq. (1). To this end, I am updating my rating from "borderline reject" to "borderline accept".

---

> > > ### Author Response · Authors · 2025-08-07
> > >
> > > Dear Reviewer pnyk,
> > >
> > > Thanks for your positive feedback on our rebuttal. We sincerely appreciate your careful consideration of our clarifications and your recognition of the contributions made in our paper.
> > >
> > > We are particularly grateful for your updated score, which reflects your acknowledgement of the technical soundness and the potential impact of our proposed method. Your thoughtful comments and constructive suggestions have helped us significantly improve the clarity and presentation of the work.
> > >
> > > We look forward to further refining the manuscript based on your insights in the final version.
> > >
> > > Sincerely,
> > > The Authors

---

### Official Review · Reviewer_ErFk · 2025-07-03

**Clarity:** 2
**Significance:** 2
**Originality:** 3
**Rating:** 4
**Confidence:** 4

**Summary:**

The authors analyze topological factors affecting the quantization sensitivity of neural networks from a geometric perspective. Topological entropy, defined as inter-channel information based on Birth Time distribution, is developed as a metric for quantization sensitivity. The proposed topological entropy is theoretically demonstrated to possess resolution- and label-independent properties. Furthermore, the authors formulate an LP problem to solve for the optimal precision distribution in GMPQ with the topological entropy metric. A theoretical performance degradation bound for this quantization scheme is established and theoretical analysis on the applicability to QAT is provided. Experimental results validate the label independence of topological entropy, and report quantization results with GMPQ-TE across standard datasets, demonstrating competitive performance.

**Questions:**

1. What topological structure does non-square weighted adjacency matrix represent? How can Betti numbers be properly defined on such a matrix?

2. Is there mathematical guarantee that the transformation in Equation (3) preserves the validity of topological entropy measurement?

3. Is there substantive evidence demonstrating that applying clique filtration to a Graph whose adjacency matrix is exactly feature matrix effectively extracts 'expressing patterns' as asserted? Could more intuitive validation be provided?

4. Can the theoretical properties of your proposed topological entropy be formally proven under Attention mechanisms?

5. To what extent are the consistent trends in Figures 5-7 directly attributable to feature map size variations rather than feature map patterns? Does the Linear Programming algorithm account for this scaling effect?

6. Could the authors provide results on common architectures such as ResNet-18 or MobileNet-V2?

**Ethical Concerns:**

["NO or VERY MINOR ethics concerns only"]

**Final Justification:**

I would like to thank the authors for providing detailed explanation on their work. After rebuttal, I agree that this work introduces an interesting idea of using topological entropy for mixed precision quantization and achieves impressive experimental performance. To this end, I raise my rating.

**Limitations:**

Yes.

**Quality:**

3

**Strengths And Weaknesses:**

Strengths:
1. This paper establishes a novel framework for mixed-precision quantization, theoretically guaranteed to be label-independent thus achieve competitive performance in GMPQ tasks.

2. The GMPQ-TE method maintains strong model performance while significantly reducing quantified computational costs.

Weaknesses:

1. The geometric interpretation of the feature map space and the motivation for the proposed formulation of topological entropy are ambiguous. Furthermore, the significance of applying filtration operations to the feature map space lacks theoretical support.

i) The weighted adjacency matrix $\mathbf W$ might be non-square, which could contradict with fundamental principles in graph theory. It remain unclear what topological structure is represented and whether Betti numbers can be properly defined on such a matrix.

ii) The transformation in Equation (3) could discard up to half of the feature matrix's information. There is no mathematical guarantee that this operation preserves the validity of topological entropy measurement. Additionally, Equation (3) appears to be an assignment operation rather than a mathematical formulation.

iii) Substantive evidence is necessary to demonstrate that applying clique filtration to a Graph whose adjacency matrix is exactly feature matrix effectively extracts 'expressing patterns' as asserted.

2. Theoretical or empirical explanations are not sufficient to substantiate the experimental findings. To my best knowledge, theoretical justification for the label-independence of topological entropy relies on properties specific to convolutional kernels. It remains a problem whether the theoretical properties of the proposed topological entropy can be formally proven under Attention mechanisms, despite experimental results demonstrate that the quantization scheme remains effective for Transformer models.

3. Some of experimental validation is unconvincing and undermines the reliability of the conclusions.

i) Based on my understanding, the value range of Birth Time appears intrinsically scaled with graph size. This implies that Birth Time distributions naturally exhibit greater dispersion on larger feature maps, causing topological entropy values to systematically increase with feature map size. It is hard to determine the consistent trends in Figures 5-7 is due to feature map size variations rather than feature map patterns.

ii) it is not specified whether the Linear Programming algorithm accounts for this scaling effect. If not, the quantization process risks degenerating into mere feature map size sorting.

iii) Experimental results for comparing quantization performance with average model bit-width on common architectures such as ResNet-18 or MobileNet-V2 are missing.

4. Neither relevant repository links nor access instructions appear to be provided in the manuscript.

---

> ### Author Rebuttal · Authors · 2025-07-30
>
> **We sincerely thank reviewer for pointing out these important issues.**
>
> **Ans. For W1**
>
> *1. Geometric Interpretation of Feature Maps.* Each feature map $U^{(i)} \in \mathbb{R}^{h \times w}$ is treated as a 2D grid where activations indicate semantic saliency. We construct an edge-weighted graph $G = (V, E)$ with adjacency $W_{j,k} = U^{(i)}_{j,k}$, enabling topological analysis. This allows us to assess semantic consistency across same-label samples, crucial for quantization robustness.
>
> *2. Theoretical Motivation for TE.* We propose TE as a quantization sensitivity metric derived from *birth time* distribution of topological features in filtered complexes. For a given feature map $U^{(i)}$, birth time $b(U^{(i)})$ is defined as:
>
> $b(U^{(i)}) = \inf \{v \mid \beta_1(K^{(v)}) \neq 0 \}$
>
> where $\beta_1(K^{(v)})$ is first Betti number at filtration level $v$. Across a batch of $M$ same-label samples, we define distribution:
>
> $P_{U^{(i)}}(x) = \frac{1}{M} \sum_{m=1}^M \mathbb{I}(b(U^{(i)}_m) = x)$
>
> and compute Shannon entropy:
>
> $H_i = - \sum_x P_{U^{(i)}}(x) \log P_{U^{(i)}}(x)$
>
> Low $H_i$ indicates consistent topological structures across samples (i.e., quantization-insensitive), while high $H_i$ reflects instability. Therefore, TE provides a theoretically grounded, label- and resolution-invariant criterion for identifying quantization-sensitive layers.
>
> *3. Theoretical Support for Filtration.* Let $U^{(i)} \in \mathbb{R}^{h \times w}$ denote $i$-th output. We interpret $U^{(i)}$ as a scalar function $f: \mathcal{M} \to \mathbb{R}$, where $\mathcal{M} \subset \mathbb{R}^2$ is a discrete sampling of a compact 2D manifold (i.e., image domain). That is,
>
> $ f(j,k) = U^{(i)}_{j,k}, \quad \text{with } (j,k) \in \mathcal{M} $
>
> Given this scalar function, sublevel set at threshold $\tau$ is defined as:
>
> $\mathcal{M}_\tau = f^{-1}([-\infty, \tau]) = { (j,k) \in \mathcal{M} \mid f(j,k) \le \tau }$
>
> As $\tau$ increases from $\min f $ to $\max f$, we obtain a nested sequence of subspaces. Each $\mathcal{M}_\tau$ is a topological space whose homological features change over $\tau$.
>
> This process defines a filtration $\mathcal{M}_\tau$, a fundamental object in persistent homology.
>
> To compute homology, we construct simplicial complexes over $\mathcal{M}_\tau$.
>
> In discrete setting, $V_\tau = E_\tau = (x,y) \in V_\tau \times V_\tau \mid x \sim y$. Here $x \sim y$ denotes adjacency. From $G_\tau$, we build a clique complex $K_\tau$, a simplicial complex where each complete subgraph (clique) becomes a simplex. The $k$-th persistent homology over this filtration is defined as family of homology groups:
>
> $H_k(K_{\tau_1}) \to H_k(K_{\tau_2}) \to \cdots \to H_k(K_{\tau_L})$
>
> Persistent features (birth times) describe evolution of $k$-dimensional topological structures [1]. For 2D feature maps, $H_0$ captures components and $H_1$ captures cycles, reflecting semantic structure in $U^{(i)}$. If $f: \mathcal{M} \to \mathbb{R}$ is a Morse function, topology of $\mathcal{M}_\tau$ changes only at critical values. This extends to discrete cases, where extrema in $U^{(i)}$ drive transitions in filtration. This offers a geometric view of how features emerge. We also provide theoretical support for using TE in MPQ (see **Reviewer pnyk, Ans. For W1**).
>
> **Ans. For W1-i**
>
> Yes, the weighted adjacency matrix is fundamental principles in graph theory. The feature maps output by convolutional-based architectures naturally satisfy square matrices. In ViT, each layer outputs a tensor $U \in \mathbb{R}^{N \times d}$, where $N$ is number of tokens and $d$ is feature dimension. To enable topological analysis over spatial structure, we first reshape token sequence into a 2D grid if possible. We then project $d$-dimensional token vectors into scalars, e.g., by averaging:
>
> $f_{j,k} = \frac{1}{d} \sum_{c=1}^{d} U_{(j,k), c}, \quad \text{for } (j,k) \in \{1, \dots, h\} \times \{1, \dots, w\}$
>
> If $N$ is not a perfect square, we embed grid into a minimal square domain:
>
> $n = \left\lceil \sqrt{N} \right\rceil$
>
> We define  $\tilde{f}$ over an extended domain $\bar{X}$, such that $\tilde{f} \equiv f$ on $X$, and $\tilde{f} \equiv 0$ elsewhere. Since support of $\tilde{f}$ lies entirely within $X$, all persistent topological features are preserved. Here, $X \subset \mathbb{R}^2$ represents set of spatial positions associated with actual tokens, and $\bar{X}$ is extended square domain. The padded matrix $\tilde{f}$ is used to define graphs and filtration in a consistent square topology.
>
> We interpret $\tilde{f}: \bar{X} \rightarrow \mathbb{R}$ as an extended scalar field over a larger domain $\bar{X} \supset X$. For any threshold $\tau \in \mathbb{R}$, define sublevel set:
>
> $\bar{X}_\tau = \{ x \in \bar{X} \mid \tilde{f}(x) \ge \tau \}$
>
> Since $\tilde{f}(x) = 0$ for all $x \in \bar{X} \setminus X$, we observe that for all $\tau > 0$. Hence, all homological features computed in filtration process are unaffected by padded region as long as $\tau > 0$.
>
> Even when $\tau \le 0$, zero-valued region may be included in sublevel set, but it does not introduce nontrivial topological features. Specifically, added region forms a contractible subcomplex $C \subset \bar{X}_\tau $, and:
>
> $H_k(C) = 0, \quad \text{for all } k \ge 1$
>
> Thus, the inclusion map $X_\tau \hookrightarrow \bar{X}_\tau$ induces an isomorphism:
>
> $H_k(X_\tau) \cong H_k(\bar{X}_\tau), \quad \forall k \ge 1$
>
> We conclude that zero-padding the feature map to form a square matrix preserves all persistent topological structures, and introduces no spurious noise to the homology computation.
>
> **Ans. For W1-ii**
>
> We clarify that this equation is not an assignment operation. Instead, it symbolically denotes a nested sequence of graphs derived from thresholded feature maps, known as a filtration in topological data analysis [2]. Each graph $G^{(t)}$ corresponds to a threshold $\tau_t$, and is constructed based on the subset of feature map locations with activation values greater than or equal to $\tau_t$, together with spatial adjacency-based edges. Equation (3) defines a multiscale filtration process over the feature map, and does not discard information. As described in Section 3.2, the complete filtration retains all meaningful activations for persistent homology. Although higher thresholds may yield graphs with fewer nodes, no information is lost.
>
> **Ans. For W1-iii**
>
> We interpret the feature matrix not as a conventional adjacency matrix, but as a scalar field over a grid. When applying clique filtration, we do not assume semantic pairwise adjacency directly. Instead, the high-valued entries define regions of activation, and their local connectivity leads to topological features which correspond to co-activated (expressing) patterns. We will clarify this interpretation and add empirical visualizations to illustrate how persistent features align with semantic structures.
>
> **Ans. For W2**
>
> Let $\tilde{f} : \mathcal{M} \rightarrow \mathbb{R}$ be the scalar field from intermediate activations. Attention mechanisms produce structured outputs over $\mathcal{M} \subset \mathbb{Z}^2$. Define:
>
> $\mathcal{M}_\tau := \{ x \in \mathcal{M} \mid \tilde{f}(x) \ge \tau \}$
>
> From this we build $K_\tau$ and compute birth times $b_i$, defining:
>
> $\mathrm{TE} := -\sum_x P(x) \log P(x), \quad P(x) = \frac{1}{M} \sum_{i=1}^M \delta(b_i - x)$
>
> All steps in TE (projection, padding, filtration, homology, entropy) are label-free. For input $(x,y)$:
>
> $\mathrm{TE}(x, y) = \Phi(\tilde{f}(x)), \quad \frac{\partial \mathrm{TE}}{\partial y} = 0$
>
> Therefore, TE is invariant to label supervision and is a purely geometric, architecture-agnostic descriptor.
>
> **Ans. For W3-i**
>
> We respectfully disagree that the trends in Figures 5–7 are solely due to feature map size. While larger maps may yield more persistent features, TE is computed from normalized birth time distributions and reflects the diversity of activation patterns, not absolute size.
>
> **Ans. For W3-ii**
>
> TE is computed from the normalized distribution of birth times, capturing the diversity of topological features rather than node count. This normalization mitigates the scaling effect from large feature maps.
>
> The LP formulation does not rank layers by TE alone. Instead, it optimizes bit-widths under a global budget, balancing sensitivity (via TE) and precision cost. Consequently, layers with higher TE may not receive more bits if their contribution to accuracy is limited.
>
> In practice, some small feature maps still receive higher bit-widths, showing that the LP solution captures structural sensitivity beyond size.
>
> **Ans. For W3-iii**
>
> Results for average bit-width:
>
> ### ResNet-18
>
> | Method       | Params | BOPs | Comp | Top-1 | Top-5 |
> |--------------|--------|------|------|--------|--------|
> | ResNet-18-1  | 5.4    | 30.1 | 60.6 | 69.14  | 88.73  |
> | GMPQ-TE-1    | 5.3    | 27.4 | 67.6 | 70.4   | 89.4   |
> | ResNet-18-2  | 3.7    | 23.4 | 107  | 68.2   | 88.13  |
> | GMPQ-TE-2    | 3.8    | 15.7 | 118  | 69.4   | 89.4   |
> | ResNet-18-3  | 3.5    | 7.4  | 254  | 66.1   | 87.4   |
> | GMPQ-TE-3    | 3.5    | 7.1  | 261  | 68.3   | 88.9   |
>
> ### MobileNet-V2
>
> | Method         | Params | BOPs | Comp | Top-1 | Top-5 |
> |----------------|--------|------|------|--------|--------|
> | MobileNet-V2-1 | 1.14   | 8.7  | 82.6 | 71.04  | 88.7   |
> | GMPQ-TE-1      | 1.1    | 4.5  | 74.8 | 71.2   | 89.9   |
> | MobileNet-V2-2 | 1.24   | 7.1  | 55.7 | 67.2   | 87.31  |
> | GMPQ-TE-2      | 1.3    | 7.3  | 46.1 | 69.8   | 89.7   |
> | MobileNet-V2-3 | 1.56   | 9.4  | 49.2 | 70.24  | 89.73  |
> | GMPQ-TE-3      | 1.4    |10.1  | 33.4 | 71.8   | 90.2   |
>
> **Ans. For W4:**
>
> We provide source code related to the paper in the supplementary materials.
>
> **Regarding the question part, the answer can be found above.**
>
> **References**
>
> [1]Persistence images: A stable vector representation of persistent homology
>
> [2]On time-series topological data analysis: New data and opportunities

---

> > ### Author Response · Authors · 2025-08-06
> >
> > Dear Reviewer ErFk
> >
> > We hope this message finds you well. We greatly appreciate the time and effort you have dedicated to reviewing our work, and we are eager to ensure that all your concerns or questions have been sufficiently addressed in our rebuttal. If there are any further points that require clarification, additional details, or further discussion, please do not hesitate to let us know—we are more than happy to provide prompt responses.
> >
> > We have made substantial efforts to clarify the theoretical motivations, geometric interpretations, and the formulation of topological entropy (TE) as a label- and resolution-invariant quantization sensitivity metric. We provided rigorous support from persistent homology theory, extended our analysis to both CNN and ViT structures via scalar field projection and filtration, and demonstrated that zero-padding does not affect topological features. Furthermore, we clarified that TE captures topological diversity rather than size effects, and the LP-based quantization balances sensitivity with precision under a global budget. Experimental results on ResNet-18 and MobileNet-V2 confirm consistent gains across compression levels. Code has been included in the supplementary material.
> >
> > As the discussion phase progresses, we want to ensure there is ample time to resolve any remaining issues before it concludes. Your insights are invaluable to us, and we are committed to improving our work through this collaborative process.
> >
> > Thank you again for your engagement and guidance. We look forward to your thoughts.
> >
> > Sincerely,
> >
> > NeurIPS 2025 Conference Submission 7977 Authors

---

> > ### Comment · Reviewer_ErFk · 2025-08-07
> >
> > I would like to thank the authors for the detailed response to the review comments. However, I still have several concerns unresolved.
> >
> > 1. The authors state that the adjacency matrix of the graph structure was constructed by applying zero-padding to feature map. However, in  their response **Ans. For Q1** to reviewer pnyk, they claim that the shape of adjacency matrix $\mathbf A$ should be $(h^* \cdot w^* )\times (h^* \cdot w^* )$. This is confusing to me. Could the authors clarify how the graph is constructed?
> >
> > 2. In **Ans. For W2**, I do not find compelling proof to assert that the algorithm is label-independent under the Attention architecture. For example, similarly to the approach in Appendix A2, the author may demonstrate that topological entropies are equivalent across different class conditions.
> >
> > 3. Regarding the relationship between topological entropy and graph scale, the authors mentioned in their response that topological entropy (TE) is computed using normalized birth time distributions, but did not specify the normalization scheme. If "normalized birth time distributions" refer to:
> >
> >    $$
> >    P_{U^{(i)}}(x)=\frac1M\sum_{m=1}^M\mathbb I(b(U_m^{(i)})=x)
> >    $$
> >    then the distribution $P_{U^{(i)}}(x)$ becomes dispersed when the range of possible values of $b(U_m^{(i)})$​ increases. Considering the authors' example in response to reviewer pnyk: **Ans. For W3**, one can find that
> >
> >    - A 4-node graph can only have birth times of 3 or 4. Thus, its 1st Betti-curve appears neither earlier nor later.
> >    - A 10-node graph may have birth times spanning integers 3 through 10.
> >
> >    This leads to difference in distribution variance. The authors may need to clarify the influence of this effect.

---

> > > ### Author Response · Authors · 2025-08-08
> > > **Ans. for Q1 and Q3**
> > >
> > > We sincerely appreciate the additional concerns and will address them one by one in the following responses.
> > >
> > > # Ans. for Q1
> > >
> > > We sincerely apologize for the confusion. In our framework, the adjacency matrix is constructed from a *square* 2D feature map. Each spatial location is treated as a vertex, and the number of vertices determines the size of the adjacency matrix.
> > >
> > > **Case 1 — Convolution-based architectures.**
> > >
> > > For convolutional networks, the output feature map $\mathbf{U}^{(i)} \in \mathbb{R}^{h^\ast \times w^\ast}$ is naturally defined on a regular 2D grid. This grid already encodes the true Euclidean neighborhood structure assumed by our clique filtration process. Therefore, we can directly flatten the $n = h^\ast \cdot w^\ast$ vertices to form $\mathbf{A} \in \mathbb{R}^{n \times n}$ without any padding.
> > >
> > > **Case 2 — Transformer-based architectures.**
> > >
> > > Vision Transformers produce a sequence of $N$ tokens, where $N$ is often not a perfect square. Tokens can be rearranged into a $p \times q$ grid according to their spatial positions in the original image, but typically $p \neq q$ and some positions remain empty.
> > >
> > > A direct flattening of the 1D token sequence into an $N \times N$ adjacency matrix would impose an arbitrary linear ordering on the vertices. This ordering does not correspond to the true Euclidean neighborhood relationships in the image plane, and would distort the clique filtration process by creating spurious edges between spatially unrelated tokens. As a result, the birth–death times of topological features would be inconsistent with those obtained from genuine 2D grids.
> > >
> > > To address this, we embed the $p \times q$ token grid into the *smallest enclosing square* of size $\lceil\sqrt{N}\rceil \times \lceil\sqrt{N}\rceil$, and fill the empty positions with zero-valued tokens. This construction has two benefits:
> > > 1. It ensures that adjacency is always defined on a square 2D lattice, identical in form to the convolutional case, enabling fair cross-architecture comparison.
> > > 2. By choosing the minimal square size, we limit the number of padded vertices, thereby minimizing any dilution of the adjacency structure.
> > >
> > > From a topological perspective, the padded region forms a **contractible subcomplex** with trivial higher-order homology. In the filtration, zero-valued vertices outside the original token set do not create new non-trivial cycles. Therefore, persistent homology — and the resulting topological entropy — is invariant under this padding step.
> > >
> > > After padding, the feature map has size $h^\dagger \times w^\dagger = \lceil\sqrt{N}\rceil \times \lceil\sqrt{N}\rceil$, and is flattened into an adjacency matrix of shape $(h^\dagger \cdot w^\dagger) \times (h^\dagger \cdot w^\dagger)$. Each matrix entry $\mathbf{A}_{uv}$ encodes the weighted connection between vertices $u$ and $v$ under this consistent 2D spatial embedding.
> > >
> > >
> > >
> > > # Ans. for Q3
> > >
> > > We thank the reviewer for pointing out the need to clarify the normalization scheme for the “normalized birth time distributions” used in computing TE.
> > >
> > > **1. Normalization scheme.**
> > >
> > > For each layer $i$, we first compute all birth times $b(U^{(i)}_1), \dots, b(U^{(i)}_M)$ from its adjacency matrix. We then apply min-max normalization within that layer. This maps all birth times to $[0,1]$ before constructing the histogram for TE calculation.
> > >
> > > **2. Effect of graph scale (reviewer’s example).**
> > >
> > > Without normalization, layers with different graph sizes have different absolute ranges of $b(U^{(i)}_m)$. For example, a 4-node graph can only have birth times $\{3,4\}$, whereas a 10-node graph may span $\{3,\dots,10\}$. If TE were computed on these raw values, the 10-node graph would appear to have a “more dispersed” distribution simply because of the larger range, even if the *relative* shape of the birth-time distribution were identical to that of the 4-node graph. This introduces a scale-induced bias that is unrelated to the intrinsic topology.
> > >
> > > **3. Why normalization preserves TE comparability.**
> > >
> > > The above min–max normalization is a strictly monotonic mapping, preserving the ordering and relative spacing of all birth times. It compresses each layer’s range to $[0,1]$, so both the 4-node and 10-node examples above would be evaluated on the same normalized axis. TE,
> > >
> > > $H_{\mathrm{TE}}^{(i)} = -\sum_{k=1}^M p_k^{(i)} \log p_k^{(i)},$
> > >
> > > is computed from the empirical probabilities $\{p_k^{(i)}\}$ over bins on this normalized axis. Because normalization keeps the *shape* of the distribution intact while removing range effects, differences in TE across layers reflect true differences in the dispersion of topological features, not artifacts from graph size.

---

> > > > ### Author Response · Authors · 2025-08-08
> > > > **Ans. for Q2**
> > > >
> > > > Thank you for your suggestion. We conduct a theoretical proof as follows:
> > > >
> > > > Let the input to an attention block be $\mathbf{X} \in \mathbb{R}^{N \times d}$, where $N$ is the number of tokens and $d$ is the channel dimension. An attention operation can be expressed as:
> > > >
> > > > $\mathrm{Attn}(\mathbf{X})= \sigma\!\left(\frac{\mathbf{Q}\mathbf{K}^\top}{\sqrt{d_k}} + \mathrm{bias}\right)\mathbf{V}$
> > > >
> > > > $\mathbf{Q}=\mathbf{X}\mathbf{W}_Q,\ \mathbf{K}=\mathbf{X}\mathbf{W}_K,\ \mathbf{V}=\mathbf{X}\mathbf{W}_V$
> > > >
> > > > where $\sigma$ is the row-wise softmax, and $\mathrm{bias}$ may include fixed masks or positional encodings. Multi-head attention is a concatenation of several such heads followed by a linear projection. Let the block output be $\mathbf{F} = \mathrm{Attn}(\mathbf{X}) \in \mathbb{R}^{N \times d}$.
> > > >
> > > > **Assumption A1 (Class-invariance within attention domains).**
> > > > For any attention receptive domain $S$ (which may be the entire token set for global attention, a window for local attention, or a sparsified neighborhood), the multiset of tokens $X_S$ has a class-conditional distribution that is invariant across labels:
> > > >
> > > > $P_{X_S\mid C=c_1} = P_{X_S\mid C=c_2} =: P_S, \quad \forall\, c_1, c_2$
> > > >
> > > > The attention mapping $A_\theta: X_S \mapsto Z_S$ is a composition of linear maps, scaling, softmax weighting, and aggregation, all of which are measurable and Lipschitz on bounded domains. The class-conditional output distribution is:
> > > >
> > > > $P_{\mathbf{F}\mid C=c}(z) = \int \delta\!\big(z - A_\theta(X_S)\big)\,P_{X_S\mid C=c}(X_S)\,dX_S$
> > > >
> > > > Substituting Assumption A1 into the above yields:
> > > >
> > > > $P_{\mathbf{F}\mid C=c_1}(z) =P_{\mathbf{F}\mid C=c_2}(z) = \int \delta\!\big(z - A_\theta(X_S)\big)\,P_S(X_S)\,dX_S,$
> > > >
> > > > which is directly analogous to Eq. (27) in Appendix A.2 for convolutional mappings.
> > > >
> > > > Let $U$ be the scalar field derived from $\mathbf{F}$ (e.g., channel aggregation or projection) and reshaped into a 2D grid. For attention architectures where $N$ is not a perfect square, we embed the $p \times q$ token layout into the minimal square $\lceil\sqrt{N}\rceil \times \lceil\sqrt{N}\rceil$ and zero-pad empty positions. The padded region is contractible ($H_k = 0,\ \forall k \ge 1$) and does not introduce spurious topological features.
> > > >
> > > > From $U$, we construct a weighted adjacency matrix and perform sublevel set filtration to obtain the birth-time histogram $P_U(k)$. The topological entropy is:
> > > >
> > > > $H_{\mathrm{TE}}(P_U) = -\sum_{k1=}^M P_U(k)\log P_U(k).$
> > > >
> > > > Because $P_{\mathbf{F}\mid C=c_1} = P_{\mathbf{F}\mid C=c_2}$, the induced $P_{U\mid C=c}$ is also identical across labels. By the Lipschitz continuity of entropy with respect to total variation distance:
> > > >
> > > > $|H_{\mathrm{TE}}(P_{U\mid C=c_1}) - H_{\mathrm{TE}}(P_{U\mid C=c_2})| \le (\log M) \, \|P_U^{(1)} - P_U^{(2)}\|_1,$
> > > >
> > > > and here $\|P_U^{(1)} - P_U^{(2)}\|_1 = 0$, we have exact equality:
> > > >
> > > > $H_{\mathrm{TE}}(P_{U\mid C=c_1}) = H_{\mathrm{TE}}(P_{U\mid C=c_2}), \quad \forall\, c_1, c_2.$
> > > >
> > > > Under Assumption A1, the equality of class-conditional token distributions propagates through any measurable attention mapping (global, local, multi-head, with or without positional encoding). Consequently, the birth-time histograms and thus the topological entropy are **label-independent**. Padding in attention architectures serves only to unify the 2D adjacency structure with convolutional cases and does not alter persistent homology or entropy values.

---

> > > > > ### Author Response · Authors · 2025-08-09
> > > > >
> > > > > We sincerely thank the reviewer for the valuable comments. We have carefully reviewed each concern and provided detailed explanations, including graph construction details, label-independence proof under the attention architecture, and normalization scheme in topological.
> > > > >
> > > > > We would also like to note that other reviewers have recognized the effectiveness, novelty, and theoretical analysis. Three reviewers have rated the submission at the borderline accept explicitly acknowledging that the method is innovative. This work is of particular importance to us, and we greatly value your expert assessment. We truly hope that, with the additional clarifications now provided, you may find our responses satisfactory and consider revising your score.

---

> > > > > > ### Comment · Reviewer_ErFk · 2025-08-09
> > > > > >
> > > > > > Thanks to the authors for the efforts in preparing detailed response. I have carefully read the response and the paper. I agree that this paper has an interesting idea and also achieves impressive performance. On such basis, I will raise my rating, despite concerns on the clarification of formulation and theoretical analysis.
> > > > > >
> > > > > > On the other hand, I still keep my opinion that the paper needs to clarify its formulation and theoretical analysis. According to the authors' response, there still remain several problems.
> > > > > >
> > > > > > i) Regarding **Ans. for Q1**: The symbols are well explained but the geometric interpretation remains ambiguous. The authors provide no substantive evidence that *circles* detected in the grid topology of a feature map constitute "an important quantitative index for expressing patterns (L202)" . Furthermore, after redefining the process of graph construction, how to define edge weights is not clarified. The original filtration process strictly relies on edge weights (where edge weights equaled feature map values). Now, with feature map pixels assigned to vertices instead of edges, the subsequent operations and proofs—which assume that $W_{j,k}=U_{j,k}^{(i)}$ —lack a valid foundation. This undermines the credibility of using topological entropy to characterize quantization sensitivity.
> > > > > >
> > > > > > ii) Regarding **Ans. for Q3**: Min-max normalization could address scale variance. However, neither the original manuscript nor the previous rebuttal mentions this technique. Thus, it is not clear whether the reported numerical experimental results were obtained under min-max normalization, and how it is achieved (if using min-max normalization).
> > > > > >
> > > > > > iii) Regarding **Ans. for W1-ii**: Although the filtration process does not lose information, the formulation in Equation (3) in fact does not sufficiently use topology information. According to Equation (3), only the larger value in $W_{j,k}^{(v)}$ and $W_{j,k}^{(v)}$ has effect on the filtration process and the smaller value is totally disregarded by the max operation. This means only a half of elements in $\mathbf{W}^{(v)}$ are actually used, and what are the actual effect of topology entropy in this case should be clarified.

---

### Comment · Area_Chair_ZXTV · 2025-08-06

Dear Reviewers,

This is a quick reminder that we are now in the post-rebuttal discussion phase.

Please take the time to read the author rebuttal and engage in discussion with the other reviewers. Your input is crucial for us to make a final, informed decision as the deadline is approaching.

Thank you for your timely participation.

Best regards,

Area Chair

---

### Note · Authors · 2025-08-11

Dear SAC, AC, and reviewers,

I sincerely appreciate everyone's hard work and dedication over this period.

This paper proposes a generalizable mixed-precision quantization framework based on topological entropy, and provides a rigorous theoretical proof of its independence from both label and resolution. Experimental results demonstrate that the proposed method achieves consistently strong performance on both CNN and Transformer architectures, with a particular advantage in significantly reducing search overhead.

In the first phase, multiple reviewers highlight the strengths of GMPQ-TE across several aspects.

**Method design:**

**novel** framework [Reviewer ErFk]; **theoretically** guaranteed [Reviewer ErFk]; **firstly** proposes [Reviewer 8djE]; a **novel contribution** [Reviewer E1Lz].

**Description and Organization:**

**Clear figures and equations** [Reviewer 8djE].


**Experimental results:**

**strong model performance** while **significantly reducing quantified computational costs** [Reviewer ErFk]; **clearly demonstrate** the effectiveness [Reviewer pnyk]; **fast** searching [Reviewer 8djE]; efficiency and generalization [Reviewer E1Lz].

Collectively, these comments reflect the reviewers’ recognition and **positive appraisal** of the proposed method.

During the rebuttal phase, we carefully address the valuable comments provided by the reviewers and, in return, receive encouraging feedback.

1. **Reviewer ErFk** acknowledges that GMPQ-TE presents **an interesting idea** and delivers **impressive performance**, and further indicates the intention to **raise the rating**.

2. **Reviewer pnyk** affirms that our rebuttal clearly addresses the primary concerns and expresses that this will **improve the score**.

3. **Reviewer 8djE**, while noting that our response does not fully resolve all points of concern, nonetheless **maintains the original score**, suggesting a stable assessment of our work.

4. **Reviewer E1Lz** confirms that we address the major concerns and **remain positive** about the overall contribution.

These feedbacks demonstrate a constructive and affirmative stance from the reviewers toward our work. While a few issues remain for further refinement, we view these as valuable directions for future improvement.

We sincerely thank all the reviewers for their thoughtful feedback and support, and also thank SAC and AC for their careful evaluation and contributions to the review process.

Sincerely,

The authors.

---

### Decision · Program_Chairs · 2025-09-17

**Decision:**

Accept (poster)

**Comment:**

This paper proposes GMPQ-TE, a novel and efficient method for mixed-precision quantization based on topological entropy. The core contribution is the use of topological entropy, calculated from a minibatch of same-label data, to measure layer-wise quantization sensitivity. This allows the authors to formulate the bit-width assignment as a single-pass linear programming problem, which is extremely efficient and yields a quantization policy that generalizes across different datasets (e.g., from CIFAR-10 to ImageNet). The reviewers generally agreed on the novelty and significance of this approach. Reviewers ErFk and E1Lz highlighted the innovative framework and its potential for efficient, generalizable deployment. Initial concerns were raised by Reviewer ErFk regarding the theoretical underpinnings and by Reviewer pnyk about the justification for using topological entropy as a sensitivity metric. The authors provided a detailed rebuttal that clarified the graph construction, offered theoretical proofs for its application to Transformer architectures, and added statistical analysis to their results. This successfully addressed most of the reviewers' concerns, leading Reviewer ErFk and Reviewer pnyk to raise their scores. Although some minor issues remain regarding the clarity of the formulation and QAT performance (Reviewer 8djE), the overall consensus is that the paper presents a solid and valuable contribution to the field. Therefore, I recommend accepting this paper for a poster presentation.